# NOVELTY DETECTION USING ENSEMBLES WITH REGULARIZED DISAGREEMENT

## ABSTRACT

Despite their excellent performance on in-distribution (ID) data, deep neural networks often confidently predict on out-of-distribution (OOD) samples that come from novel classes instead of flagging them for expert evaluation. Even though conventional OOD detection algorithms can distinguish far OOD samples, current methods that can identify near OOD samples require training with labeled data that is very similar to these near OOD samples. In turn, we develop a new ensemble-based procedure for *semi-supervised novelty detection* (SSND) that only utilizes a mixture of unlabeled ID and OOD samples to achieve good detection performance on near OOD data. It crucially relies on regularization to promote diversity on the OOD data while preserving agreement on ID data. Extensive comparisons of our approach to state-of-the-art SSND methods on standard image data sets (SVHN/CIFAR-10/CIFAR-100) and medical image data sets reveal significant gains with negligible increase in computational cost [1].

## 1 INTRODUCTION

Despite achieving great in-distribution (ID) performance, deep neural networks (DNN) often have trouble dealing with samples in the test set that are out-of-distribution (OOD), i.e. test inputs that are unlike the data seen during training. For example, DNNs often make incorrect predictions with high confidence when new unseen classes emerge over time (e.g. undiscovered bacteria Ren et al. (2019), new diseases Katsamenis et al. (2020)), or when data suffers from distribution shift (e.g. corruptions Lu et al. (2019), environmental changes Kumar et al. (2020)).

If the labels of these OOD samples follow the same conditional probability distribution as the training set Shimodaira (2000), it is most desirable to output calibrated uncertainty estimates in addition to a single predicted value, as in Neal (1996); Gal & Ghahramani (2016); Malinin & Gales (2018); Lakshminarayanan et al. (2017). In contrast, when this assumption does not hold and some inputs might come from previously unseen classes, we would like to *detect* such novel samples and bring them to the attention of experts. More specifically, in *novelty detection*, we want to detect inputs $x$ that satisfy $P_X(x) < \alpha$ for a small threshold $\alpha$, where $P_X$ is the marginal training (ID) distribution.

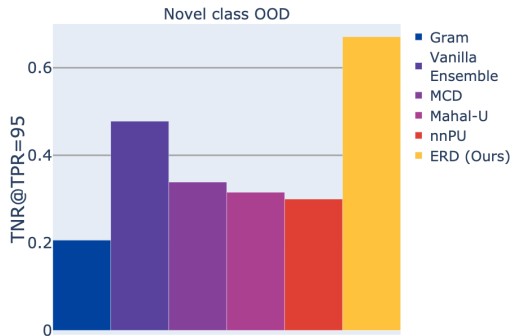

Figure 1: Comparison of methods that are applicable in the SSND setting. Our method shows better performance on challenging near OOD data sets. Averages over the data sets presented in Table 2.

Previous works that have tackled this problem are known as open-set recognition and OOD detection methods. In what follows, we use the terms novelty detection (ND) and OOD detection interchangeably. Related problems such as aforementioned uncertainty estimation or one-class (OC) classification have slightly different objectives but also include methods that are applicable to novelty detection.

---

[1]Our code is publicly available at https://bit.ly/3a7aQyN

Table 1: Taxonomy of novelty detection methods, categorized according to data availability (**horizontal axis**) and overall objective (**vertical axis**). We highlight the ensemble-based methods.

| | P-UND | SND | Different OOD A-UND | Synthetic OOD A-UND | SSND | UND |
|---|---|---|---|---|---|---|
| Learn $P_X$ | [RH21] | [GKRB13, DKT19, RVGBM20] | | OC classif. [SLYJP21, TMJS20] | PU [KNPS17] | Generative e.g. [AAB18], OC classif. e.g. [SPSSW01] |
| Learn $P_X$ using $y$ | ViT [FRL20] | Mahalanobis [LLLS18], MCD [YA19] | | Contrastive loss [TMJS20, LA20] | SSND for shallow models [MBGBC10, BLS10], U-LAC [DYZ14, ZZMZ20] | Gram [SO19], Contrastive [WBRSN20], OpenHybrid [ZLGG20] |
| Uncertainty of $P_{Y\|X}$ | | ODIN [LLS18] | DPN [MG18], Outlier Exposure [HMD19] | Calibrate using GAN images [LLLS18], noise [HTLID19] or uniform samples ([JLMG20]) | — | Bayesian methods e.g. [GG16], Vanilla Ensemble [LPB17] |

Numerous novelty detection methods are successful for simple benchmarks where the OOD samples are far from the training samples (such as SVHN vs CIFAR10). As Winkens et al. (2020) and a number of concurrent works (Tack et al., 2020; Fort et al., 2021) recently noted, these methods however have subpar performance on near OOD data, for instance when OOD samples are drawn from unseen classes from the same data set (e.g. CIFAR100 vs CIFAR10). A recent method that is reported to have high near OOD detection performance involves tuning large models pretrained on ImageNet21k (Fort et al., 2021) and, hence, arguably relies on using during training OOD data that contains classes that are close to the unseen CIFAR classes during test time. However, in many scientific applications that use, for instance, medical or satellite images, such large data sets for pretraining are not available. In particular, truly novel classes will be inherently dissimilar from any previously available data while still sharing some of the same characteristics.

Instead, a more realistic scenario assumes access to a small batch of unlabeled test data that includes ID and OOD samples without knowing which are the outliers. We can use this set during training to ultimately perform well on future samples (i.e. semi-supervised novelty detection (SSND) (Blanchard et al., 2010)) or to detect outliers from that particular test batch itself (reminiscent of transductive OOD detection (Scott & Blanchard, 2008)). Even though using unlabeled data has the potential to improve detection performance as argued in Scott & Blanchard (2009), existing SSND methods for deep neural networks (Kiryo et al., 2017; Yu & Aizawa, 2019; Guo et al., 2020; Zhang et al., 2020b) do not leverage the unlabeled set well enough to improve near OOD detection performance, as shown in Figure 1. This includes previous methods that try to obtain a diverse ensemble and use it for novelty detection, such as MCD (Yu & Aizawa, 2019).

In this paper, we introduce Ensembles with Regularized Disagreement (ERD), an ensemble method for SSND that successfully takes advantage of the unlabeled data to achieve the right amount of disagreement between the model predictions. The models in our ensemble are trained to not only fit the training data but also the artificially labeled samples from the unlabeled set, while aiming to achieve good validation accuracy. In particular,

- we argue why regularizing disagreement is crucial for OOD detection with ensemble methods.

- we give a justification, backed by theoretical arguments, as to why training with early stopping and with artificial labels assigned to the unlabeled set achieves the right amount of disagreement.

- we test our method on many near OOD tasks, including medical data, demonstrating significant gains with a negligible increase in computation cost compared to vanilla ensembles.

## 2 RELATED WORK AND A TAXONOMY OF OOD DETECTION METHODS

In this section we give an overview of approaches that are related to ours and categorize them with respect to 1) data availability and 2) the final objective. Further, we list a few representative approaches in Table 1 and refer the reader to surveys such as Bulusu et al. (2020) for a thorough literature overview.

### 2.1 TAXONOMY ACCORDING TO DATA AVAILABILITY

In this section we present methods that are ordered by a decreasing necessary amount of labeled OOD data or data similar to OOD data at test time.

**Supervised and augmented unsupervised ND.** We call methods that use labeled OOD samples from the test distribution for calibration or hyperparameter tuning *supervised ND* (SND) (Yu & Aizawa, 2019; Lee et al., 2018; Liang et al., 2018; Ruff et al., 2020). Alternatively, some works use during training either real, known outliers Hendrycks et al. (2019); Malinin & Gales (2018) or OOD-like synthetically generated data to simulate the test OOD distribution, in what we call the *augmented unsupervised ND* (A-UND). However, the performance of these methods relies on proxy OOD data that is similar to test OOD. Concurrent works also propose to use models large-scale models that are pretrained on datasets that contain classes similar to the novel classes (Fort et al., 2021; Reiss & Hoshen, 2021), referred to as *pretrained unsupervised ND*.

**Semi-supervised novelty detection (SSND, ours).** This setting is the one we study in this work.[2] We assume that, apart from class-labeled training data, we also have access to a small batch of unlabeled data drawn from the test distribution. It consists of ID and OOD data *without explicitly knowing* which samples are ID and OOD. SSND methods attempt to detect OOD samples from a new batch drawn from the same test distribution (Blanchard et al., 2010; Mũnoz-Marí et al., 2010; du Plessis et al., 2014; Liu et al., 2018). This is similar to the goal of approaches that use unlabeled data for learning with augmented classes (U-LAC) (Da et al., 2014; Guo et al., 2020; Zhang et al., 2020b). The SSND setting is related to transductive novelty detection (Scott & Blanchard, 2008; Guo et al., 2020), where the test set coincides with the unlabeled set used for training.

**Unsupervised novelty detection (UND).** In this less restrictive setting, only ID data is available during training (Lakshminarayanan et al., 2017; Sastry & Oore, 2019; Nalisnick et al., 2019; Choi et al., 2018), which generally leads to a poorer performance. Even though Scott & Blanchard (2009) prove for shallow models that unlabeled data available in SSND should significantly improve detection, it remains unclear how well deep neural networks can leverage the unlabeled data set effectively.

## 2.2 TAXONOMY ACCORDING TO OVERALL OBJECTIVE

We now discuss the different objectives of methods in the literature that can be used to detect OOD samples and explicitly list those that do not explicitly mention this keyword.

**Learning the ID marginal $P_X$.** Since we define OOD samples as all $x$ for which $P_X(x) < \alpha$, if we had access to the marginal training distribution $P_X$, we would have perfect OOD detection for any $x \notin \mathcal{X}_{ID}$. Generative models (Choi et al., 2018; Akçay et al., 2018; Nalisnick et al., 2019), however, struggle to learn the density from finite samples when the data is high-dimensional. Alternatively, one-class classification (Mũnoz-Marí et al., 2010; Ruff et al., 2020; Sohn et al., 2021) and PU learning approaches (du Plessis et al., 2014; Kiryo et al., 2017) try to directly learn a discriminator between ID and OOD data, but tend to produce indistinguishable representations for inliers and outliers when the ID distribution consists of many diverse classes.

**Learning $P_X$ using label information. (ours)** When the ID training set has class labels, one can take advantage of that additional information to distinguish points in the support of $P_X$ from OOD data, for instance, by using the intermediate representations of trained neural networks (Lee et al., 2018; Sastry & Oore, 2019). Often the task is to also simultaneously predict well on ID data, known in the literature as the problem open-set recognition (Zhang et al., 2020a; Geng et al., 2021).

**Learning uncertainty estimates for $P_{Y|X}$.** Uncertainty estimates optimized for minimal calibration error can naturally be used for OOD samples. Many uncertainty quantification methods are based on a Bayesian framework (Gal & Ghahramani, 2016; Malinin & Gales, 2018; Graves, 2011; Blundell et al., 2015) or calibration improvement (Liang et al., 2018; Hafner et al., 2019) – neither of them however perform as well as other OOD methods mentioned above (Ovadia et al., 2019).

## 3 PROPOSED METHOD

In this section we introduce our proposed algorithm, ERD, and provide a principled justification for the key ingredients that lead to the improved performance of our method.

---

[2] We use the same definition of SSND as the survey by Bulusu et al. (2020), whereas some works use the term to refer to SND (Gornitz et al., 2013; Daniel et al., 2019; Ruff et al., 2020) or UND (Song et al., 2017; Akçay et al., 2018) according to our taxonomy.

## 3.1 THE COMPLETE ERD PROCEDURE

Recall that we have access to both a labeled training set $S = \{(x_i, y_i)\}_{i=1}^{n} \sim P$, where $x_i \in \mathcal{X}$ are covariates and $y_i \in \mathcal{Y}$ are discrete labels, and an unlabeled set $U$, which contains both ID and unknown OOD samples. Moreover, we initialize the models of the ensemble using weights pretrained on $S$.[3]

| **Algorithm 1:** Fine-tuning the ERD ensemble |
| --- |
| **Input:** Train set $S$, ID Validation set $V$, Unlabeled set $U$, Model $\tilde{f}$ pretrained on $S$, Ensemble size $K$ |
| **Result:** ERD ensemble $\{f_{y_i}\}_{i=1}^{K}$ |
| Sample $K$ different labels $\{y_1, ..., y_K\}$ from $\mathcal{Y}$ |
| **for** $c \leftarrow \{y_1, ..., y_K\}$ **do** // fine-tune $K$ models |
| $\quad$ $f_c \leftarrow Initialize(\tilde{f})$ |
| $\quad$ $(U, c) \leftarrow \{(x, c) : x \in U\}$ |
| $\quad$ $f_c \leftarrow RegularizedFineTuning(f_c, S \cup (U, c); V)$ |
| **return** $\{f_{y_i}\}_{i=1}^{K}$ |

| **Algorithm 2:** OOD detection with ERD |
| --- |
| **Input:** Ensemble $\{f_{y_i}\}_{i=1}^{K}$, Test set $T$, $O = \emptyset$, Threshold $t_0$, Disagreement metric $\rho$ |
| **Result:** $O$, i.e. the OOD elements of $T$ |
| **for** $x \in T$ **do** // run hypothesis test |
| $\quad$ **if** $(Avg \circ \rho)(f_{y_1}, ..., f_{y_K})(x) > t_0$ **then** |
| $\quad\quad$ $O \leftarrow O \cup \{x\}$ |
| $\quad$ **else** |
| $\quad\quad$ predict $x$ using $\tilde{f}$ |
| **return** $O$ |

The entire training procedure is described in Algorithm 1. For training a single model in the ensemble, we assign a label $c \in \mathcal{Y}$ to all the unlabeled samples in $U$, resulting in the $c$-labeled set that we denote as $(U, c) := \{(x, c) : x \in U\}$. We then fine-tune a classifier $f_c$ on the union $S \cup (U, c)$ of the correctly-labeled training set $S$, and the unlabeled set $(U, c)$. The model $f_c$ that we output is regularized to have high validation accuracy on the validation set while still achieving a low training error on $S \cup (U, c)$. We create an ensemble of $K$ classifiers $f_c$ by choosing a different artificial label $c \in \mathcal{Y}$ for every model[4]. Finally, during test time in Algorithm 2, any samples with an aggregate disagreement metric (described in Section 3.2) larger than a threshold value $t_0$ are flagged as OOD while the rest can be predicted using the initial model trained on the labeled training set.

Intuitively, the training procedure encourages the models to produce different predictions on the OOD samples in $U$, while regularization prevents them from fitting the incorrect label $c$ and hence disagreeing on the ID points. We argue in Section 3.2 why regularized disagreement is essential and in Section 3.3 why we can find such models using early stopping.

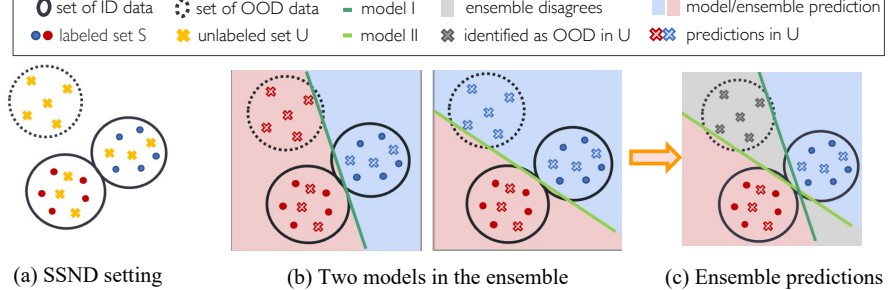

| (a) SSND setting | (b) Two models in the ensemble | (c) Ensemble predictions |

Figure 2: **Left:** Sketch of the SSND setting. **Middle and Right:** OOD detection with a diverse ensemble.

## 3.2 ENSEMBLES WITH REGULARIZED DISAGREEMENT

We now discuss how we can use ensembles with disagreement detect OOD samples and why the right amount of diversity is crucial. Note that we can cast the OOD detection problem as a hypothesis test with null hypothesis $H_0 : x \in \mathcal{X}_{ID}$. Our Algorithm 2 tests the null hypothesis by using an ensemble-based disagreement score: The null hypothesis is *rejected* and we report $x$ as OOD (*positive*) if the score is larger than a threshold $t_0$ (Section 4.3 elaborates on the choice of $t_0$). Ideally, the test should have high power (flag true OOD as OOD) and low false positive rate (avoid flagging true ID as OOD).

For simplicity of illustration, let us assume a training set with binary labels and a semi-supervised novelty detection setting as depicted in Figure 2 a). If we obtain two different models as in Figure 2

---

[3]In Section 4 we also present a version of ERD trained from random initializations, i.e. ERD++.

[4]Choosing the ensemble size $K = 5$ is sufficient for all of our experiments even for $|\mathcal{Y}| \sim 50$ or larger.

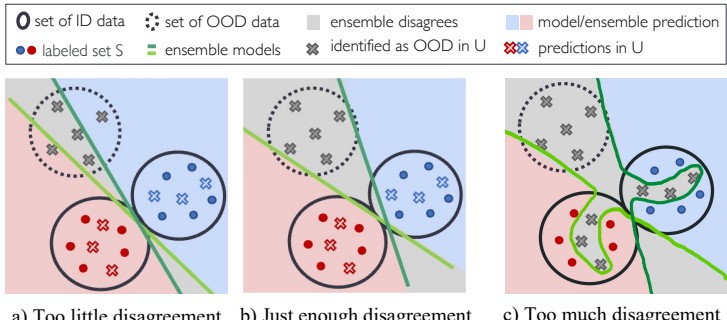

a) Too little disagreement  b) Just enough disagreement  c) Too much disagreement

Figure 3: Varying degrees of ensemble disagreement and how that influences what regions are flagged OOD.

b), the ensemble *agrees* on the blue and red area and *disagrees* on the gray area depicted in Figure 2 c). If we flag as OOD all samples in the region where the models disagree, in this scenario we have high power and a low false positive rate. To extend the concept of disagreement for a multi-class model and multi-model ensemble, we propose to use the average disagreement between the softmax outputs of the $K$ models in the ensemble

$$(\text{Avg} \circ \rho)(f_1(x), ..., f_K(x)) := \frac{2}{K(K-1)} \sum_{i \neq j} \rho\left(f_i(x), f_j(x)\right),$$ (1)

where $\rho$ is a measure of disagreement between the softmax outputs of two predictors, for example the total variation distance $\rho_{\text{TV}}(f_i(x), f_j(x)) = \frac{1}{2}\|f_i(x) - f_j(x)\|_1$ used in our experiments[5]. We provide a thorough discussion on the soundness of this statistic for disagreeing models and compare it with previous metrics in Appendix B.

We would like to emphasize that the two models in Figure 2 are *just diverse enough* to obtain both high power and low false positive rate at the same time. In particular, previous ensemble methods usually do not reach the "sweet spot": either they have too little disagreement as in Figure 3 a), resulting in low power, or they disagree too much, resulting in high false positive rate as in Figure 3 c). In these figures, one can easily see that at the the sweet spot of what we call *regularized disagreement* in Figure 3 b), the models are quite diverse while maintaining high ID validation accuracy. Supported by previous empirical and theoretical observations, we now argue that we can find the sweet spot via early-stopped fine-tuning on the artifically labeled unlabeled set.

### 3.3 Disagreement via artificial labels and regularization via early stopping

In this section we show how we can obtain a neural network that achieves the right amount of disagreement using unlabeled data and regularization via *early stopping* (refer to the following theoretical works on early stopping implicitly restricting model complexity Yao et al. (2007); Raskutti et al. (2013); Wei et al. (2017)). Previous methods have also tried to leverage a set of unlabeled data to obtain more diverse ensembles for OOD detection (Yu & Aizawa, 2019; Jain et al., 2020) or for better predictive performance(Bennett et al., 2002; Zhang & Zhou, 2010). The Maximum discrepancy method (MCD), the only candidate that can be used for our large-scale datasets, tends to result in ensembles that do not disagree enough on OOD data, resulting in subpar performance (see Figure 9a in Appendix B).

We argue that Algorithm 1 when regularized via early stopping succeeds based on two observations in the literature: empirically, neural networks can fit arbitrary labels perfectly (Zhang et al., 2016) while on the other hand, noisy samples with incorrect labels are often fit after the correctly labeled samples Yilmaz & Heckel (2019); Li et al. (2020); Song et al. (2020); Liu et al. (2020); Xia et al. (2021). This motivates our procedure that 1) fine-tunes models to fit artificial labels $c$ during training, while we 2) use the model at an earlier stopping time that achieves high validation accuracy while enough of the samples in the $U$ have been fit. We now give a more detailed argument why such a stopping time can lead to good regularized disagreement.

Recall that in our approach, in addition to the correct labels of the ID training set $S$, each member of the ensemble tries to fit one label $c$ to the entire unlabeled set $U$ that can be further partitioned into

---

[5]We also expect other distance metrics to be similarly effective.

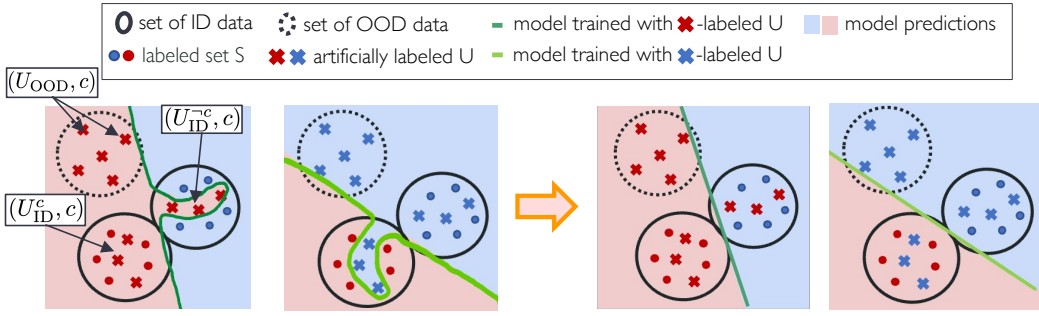

a) Two models trained to fit artifically labeled data      b) Effect of regularization on trained models

Figure 4: Regularization prevents individual models trained on $S \cup (U, c)$ from fitting $(U_{\mathrm{ID}}^{\neg c}, c)$.

$$(U, c) = (U_{\mathrm{ID}}, c) \cup (U_{\mathrm{OOD}}, c) = \{(x, c) : x \in U_{\mathrm{ID}}\} \cup \{(x, c) : x \in U_{\mathrm{OOD}}\},$$

where $U_{\mathrm{ID}} := U \cap \mathcal{X}_{ID}$ and $U_{\mathrm{OOD}} := U \setminus U_{\mathrm{ID}}$. Moreover, assuming that the labels of an ID input $x$ is deterministically $y^*(x)$, we can partition the set $(U_{\mathrm{ID}}, c)$ (see Figure 4a) into a subset of effectively "correctly labeled" samples $(U_{\mathrm{ID}}^{c}, c)$ and "incorrectly labeled" samples $(U_{\mathrm{ID}}^{\neg c}, c)$:

$$(U_{\mathrm{ID}}^{\neg c}, c) := \{(x, c) : x \in U_{\mathrm{ID}} \text{ with } y^*(x) \neq c\}$$
$$(U_{\mathrm{ID}}^{c}, c) := \{(x, c) : x \in U_{\mathrm{ID}} \text{ with } y^*(x) = c\}.$$

Note that $(U_{\mathrm{ID}}^{\neg c}, c)$ can be viewed as the noisy samples of our entire training set. Assuming the OOD samples are representative, if the models fit $(U_{\mathrm{OOD}}, c)$ perfectly to different artificial labels $c$, then they will disagree on the OOD data in the labeled set, as well as on unseen similar OOD samples, leading to a diverse ensemble. However, if the models also fit the samples in $(U_{\mathrm{ID}}^{\neg c}, c)$, the ensemble becomes too diverse, as shown in Figure 4 a). As noted before, early stopping prevents models from fitting training samples with label noise, which, in our case, amounts to the incorrectly labeled subset $(U_{\mathrm{ID}}^{\neg c}, c)$. In Proposition A.1 in Appendix A we provide a rigorous proof based on previous theoretical results that establishes the existence of such an optimal stopping time under mild assumptions on the data distributions.

To find the best stopping time in practice, we use a validation set of labeled ID points to select an intermediate checkpoint before convergence. As a model starts to fit $(U_{\mathrm{ID}}^{\neg c}, c)$, i.e. the wrongly labeled ID samples in $U_{\mathrm{ID}}$, it also predicts the label $c$ on some validation ID points, leading to a decrease in validation accuracy, as shown in Figure 5. In our experiments, we wait for one epoch to allow for the fine-tuning to have any effect at all, and then pick the iteration with the largest validation accuracy (indicated by the vertical line in the figure). In Appendix I we show that the trend depicted in Figure 5 persists for various different data sets.

Finally, we note that instead of early stopping we could also explicitly regularize using dropout or weight decay. However, running a grid search to select the right hyperparameters can be more computationally expensive than simply using one run of the training process to select the optimal stopping time.

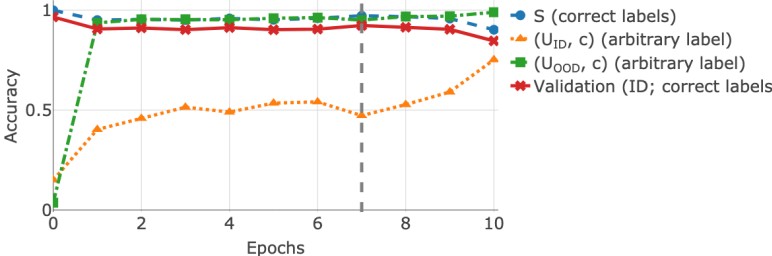

Figure 5: Accuracy during fine-tuning a model pretrained on $S$ (epoch 0 indicates values obtained with the initial pretrained weights). The samples in $(U_{\mathrm{OOD}}, c)$ are fit first, while the model reaches high accuracy on $(U_{\mathrm{ID}}, c)$ much later. We fine-tune for at least one epoch and then early stop when the validation accuracy starts decreasing after 7 epochs (vertical line). The model is trained on SVHN[0:4] as ID and SVHN[5:9] as OOD.

## 4 EXPERIMENTAL RESULTS

In this section we evaluate the OOD detection performance of ERD for deep neural networks on several image data sets. We find that our approach outperforms all baselines on difficult OOD detection scenarios. In addition, we discuss some of the trade-offs that impact ERD's performance.

### 4.1 DATA SETS

We report results on near and far OOD detection scenarios using standard data sets and a recent OOD detection benchmark for medical images Cao et al. (2020) (see Appendix D for details):

**Easy/Far OOD data.** ID and OOD samples come from strikingly different data sets (e.g. CIFAR10 or CIFAR100 as ID and SVHN as OOD). These are the settings considered in the majority of the literature and on which most baselines perform well.

**Hard/Near OOD data.** The OOD data consists of "novel" classes that resemble the ID samples. For the standard image data sets we consider half of the classes as ID, and the other half as OOD. For the medical benchmarks near OOD data consists of unseen diseases. The similarities between the ID and the OOD classes make these settings significantly more challenging.

For all scenarios, we used a labeled training set (e.g. 40K samples for CIFAR10), a validation set with ID samples (e.g. 10K samples for CIFAR10) and an unlabeled test set where half of the samples are ID and the other half are OOD (e.g. 5K ID samples and 5K OOD samples for CIFAR10 vs SVHN). For evaluation, we use a holdout set containing ID and OOD samples in the same proportions as the unlabeled set. In Appendix E.1 we show that ERD can also successfully identify the outliers from the unlabeled set used for fine-tuning. Furthermore, in Appendix E.4 we present results obtained with a smaller unlabeled set of only 1K samples.

### 4.2 BASELINES

We compare our method against a wide range of baselines that require different access to OOD data for training, as indicated in Table 1.

**Semi-supervised novelty detection.** We primarily compare our method to approaches that can be used in the semi-supervised setting for deep neural networks, in which a small set of unlabeled ID and OOD samples is available. The MCD method Yu & Aizawa (2019) trains an ensemble of two classifiers with different types of predictive distributions on the unlabeled samples: one model gives high-entropy predictions, while the other has low entropy. Notably, this method uses oracle OOD data for hyperparameter tuning. Furthermore, positive-unlabeled (PU) learning du Plessis et al. (2014) considers a binary classification setting, in which the labeled data comes from one class (i.e. ID samples, in our case), while the unlabeled set contains a mixture of samples from both classes. Crucially, PU learning methods, like nnPU Kiryo et al. (2017), require oracle knowledge of the ratio of OOD samples in the unlabeled set.

In addition to these methods, we propose two more baselines that use an unlabeled set. Firstly, we present a version of the Mahalanobis approach (*Mahal-U*) that is calibrated using the unlabeled set. Secondly, since PU learning requires access to the OOD ratio of the unlabeled set, we also consider a less burdensome alternative: a *binary classifier* trained to separate the training data from the unlabeled set and regularized with early stopping like our method.

**Unsupervised novelty detection.** When it comes to methods that use *no OOD* data for training, the current SOTA on the usual benchmarks is the Gram method Sastry & Oore (2019). Other approaches that use no OOD data include vanilla ensembles Lakshminarayanan et al. (2017), deep generative models that tend to give undesirable results for OOD detection Kirichenko et al. (2020), or various Bayesian approaches that are often poorly calibrated on OOD data Ovadia et al. (2019).

**Other methods.** We also compare with Outlier Exposure Hendrycks et al. (2019) and Deep Prior Networks (DPN) Malinin & Gales (2018) which use TinyImages as known outliers during training, irrespective of the OOD set used for evaluation (A-UND). On the other hand, the Mahalanobis baseline Lee et al. (2018) is tuned on samples from the same OOD distribution used for evaluation (SND). Preliminary analyses revealed that generative models and one-class classification methods perform poorly on near OOD data sets (see also Appendix E.2).

**Choice of hyperparameters.** For all the baselines, we use the default hyperparameters suggested by their authors on the same ID data set. The Gram method requires laborious hyperparameter tuning for multi-layer perceptron (MLP) models, so we do not consider it for the MNIST and FMNIST data

Table 2: AUROC and TNR@95 for different OOD detection scenarios (the numbers in squared brackets indicate the ID or OOD classes). We highlight the **best ERD** variant and *best baseline*. The asterisk marks baselines proposed in this paper. nnPU and MCD ($^\dagger$) use oracle information about the OOD data.

| ID data | OOD data | Other settings | | | | | Unknown OOD | | | | |
|---|---|---|---|---|---|---|---|---|---|---|---|
| | | Vanilla Ensembles | Gram | DPN | OE | Mahal. | nnPU$^\dagger$ | MCD$^\dagger$ | Mahal-U | Bin. Classif. * | ERD * |
| | | AUROC ↑ / TNR@95 ↑ | | | | | | | | | |
| SVHN | CIFAR10 | 0.97 / 0.88 | 0.97 / 0.86 | *1.00 / 1.00* | *1.00 / 1.00* | 0.99 / 0.98 | *1.00 / 1.00* | 0.97 / 0.85 | 0.99 / 0.95 | 1.00 / 1.00 | **1.00 / 0.99** |
| CIFAR10 | SVHN | 0.92 / 0.78 | *1.00* / 0.98 | 0.95 / 0.85 | 0.97 / 0.89 | 0.99 / 0.96 | *1.00 / 1.00* | *1.00* / 0.98 | 0.99 / 0.96 | 1.00 / 1.00 | **1.00 / 1.00** |
| CIFAR100 | SVHN | 0.84 / 0.48 | 0.99 / 0.97 | 0.77 / 0.44 | 0.82 / 0.50 | 0.98 / 0.90 | *1.00 / 1.00* | 0.97 / 0.73 | 0.98 / 0.92 | 1.00 / 1.00 | **1.00 / 1.00** |
| FMNIST [0,2,3,7,8] | FMNIST [1,4,5,6,9] | 0.64 / 0.07 | – / – | 0.77 / 0.15 | 0.66 / 0.12 | 0.77 / 0.20 | *0.95 / 0.71* | 0.78 / 0.30 | 0.82 / 0.39 | 0.95 / 0.66 | **0.94 / 0.67** |
| SVHN [0:4] | SVHN [5:9] | 0.92 / 0.69 | 0.81 / 0.31 | 0.87 / 0.19 | 0.85 / 0.52 | 0.92 / 0.71 | *0.96 / 0.73* | 0.91 / 0.51 | 0.91 / 0.63 | 0.81 / 0.40 | **0.95 / 0.74** |
| CIFAR10 [0:4] | CIFAR10 [5:9] | 0.80 / 0.39 | 0.67 / 0.15 | *0.82* / 0.32 | *0.82 / 0.41* | 0.79 / 0.27 | 0.61 / 0.11 | 0.69 / 0.25 | 0.64 / 0.13 | 0.85 / 0.43 | **0.93 / 0.70** |
| CIFAR100 [0:49] | CIFAR100 [50:99] | *0.78 / 0.35* | 0.71 / 0.16 | 0.70 / 0.26 | 0.74 / 0.31 | 0.72 / 0.20 | 0.53 / 0.06 | 0.70 / 0.26 | 0.72 / 0.19 | 0.66 / 0.13 | **0.82 / 0.44** |
| Average | | 0.84 / 0.52 | 0.86 / 0.57 | 0.84 / 0.46 | 0.84 / 0.54 | *0.88* / 0.60 | 0.86 / *0.66* | 0.86 / 0.55 | 0.86 / 0.60 | 0.89 / 0.66 | **0.95 / 0.79** |

sets.[6] For the binary classifier and nnPU, we pick hyperparameters only to optimize the loss on an ID validation set. We defer the details regarding training the models to Appendix C.

## 4.3 OUR METHOD AND EVALUATION

**ERD.** We present results for two flavors of our method. For one, we fine-tune each model in the ensemble with early stopping, starting from weights that are pretrained on the labeled ID set $S$ (ERD). We also train the models from random initializations (ERD++) that can obtain slightly better OOD detection at the cost of more training iterations. We train ensembles of $K = 5$ MLP models for MNIST and FMNIST and ResNet20 He et al. (2016) networks for the other settings (results for other architectures and ensemble sizes are presented in Appendix E.7 and E.8, respectively). We choose the arbitrary label assigned to the unlabeled set at random, without replacement (see Appendix E.8 for a discussion on the impact of the choice of arbitrary label). For each model in the ensemble we perform post-hoc early stopping: we train for 10 epochs (100 epochs for ERD++) and select the iteration with the lowest validation loss. The other hyperparameters for training are chosen to maximize validation accuracy on the ID data.

**Evaluation.** As in standard hypothesis testing problems, choosing different thresholds for rejecting the null hypothesis leads to different false positive and true positive rates (FPR and TPR, respectively). The ROC curve follows the FPR and the TPR for all possible threshold values and the area under the curve (AUROC; larger values are better) captures the performance of a statistical test without having to select a specific threshold. In addition to the AUROC, we also use the TNR at a TPR of 95% (TNR@95; larger values are better) for evaluation.[7]

## 4.4 MAIN RESULTS

Table 2 summarizes the main empirical results. On the easy scenarios (top part of the table) most methods achieve near-perfect OOD detection with AUROC close to 1. However, on the novelty detection scenarios (bottom part), ERD has a clear edge over other baselines, even when they are calibrated on *oracle OOD* data, or when they use the true OOD ratio of the unlabeled set, e.g. nnPU. The substantial gap between ERD and other approaches, both in average AUROC and average TNR@95, indicates that our method lends itself well to practical situations when accurate OOD detection is critical. We note that repeated runs of ERD show a small variance $\sigma^2 < 0.01$ in the detection metrics. In Appendix E we show that our method successfully identifies OOD samples with mild distribution shift (e.g. corrupted CIFAR10 Hendrycks & Dietterich (2019), CIFAR10v2 Recht et al. (2018), ObjectNet Barbu et al. (2019)), which provides further evidence that ERD is well-suited for the most difficult of OOD detection tasks.

For the medical OOD detection benchmark we show in Figure 6a the average AUROC achieved by some representative baselines taken from Cao et al. (2020). Our method improves the average

---

[6]We note that the code provided by the authors does not include the configurations required for MLP models.

[7]In practice, choosing a good rejection threshold is important. A recent work Liu et al. (2018) proposes a criterion for setting the threshold that is tailored specifically to the SSND setting. Alternatively, one can choose the threshold so as to achieve a desired FPR, which we can estimate using a validation set of ID samples.

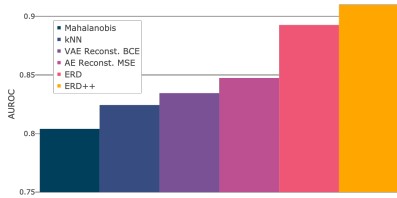

(a) OOD detection performance on medical data

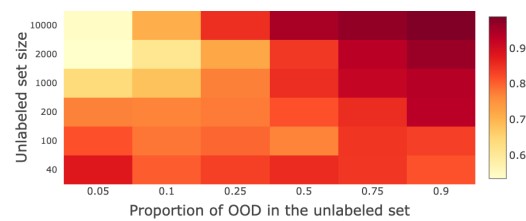

(b) Effect of OOD proportion on detection

Figure 6: **Left:** AUROC averaged over all scenarios in the medical OOD detection benchmark. The values for the baselines are computed using the code from Cao et al. (2020). **Right:** The AUROC of ERD as the number and proportion of ID (CIFAR10[0:4]) and OOD (CIFAR10[5:9]) samples in the unlabeled set are varied.

AUROC from $0.85$ to $0.91$, compared to the best baseline. We refer the reader to Cao et al. (2020) for precise details on the methods. Appendix F contains more results, as well as additional baselines.

## 5 CONCLUSION

In this section, we discuss the possible limitations of using ERD in practice before summarizing the advantages compared to other methods and outlining potential avenues for future work.

**Discussion on potential limitations of ERD and SSND.** First, as mentioned in Section 3.3, ERD requires the OOD samples in the unlabeled set to be representative of the OOD distribution. Even though this may not be realistic for anomaly detection where outliers are rare, it may very well be feasible when a novel class (such as a disease like COVID-19) emerges at inference time. We empirically investigate the impact on the performance of ERD of the size and the ratio of OOD samples in the unlabeled set ($\frac{|U_{\text{OOD}}|}{|U_{\text{ID}}|+|U_{\text{OOD}}|}$). We find that there is a broad spectrum of values for which ERD maintains a good performance, as indicated in Figure 6b (see also Appendix H). Moreover, in Appendix I and J we provide insight as to why near OOD data affects the performance of ERD.

Another limitation concerns the general applicability of SSND: the OOD data in the unlabeled set used for fine-tuning needs to match the OOD data at test time. In the following we argue using an example that i) it may be inherently valuable to predict the OOD samples in the unlabeled set itself; and ii) ERD allows for a different test OOD distribution, at the cost of a slight delay.

Consider, for instance, a medical center that uses an automated system for real-time diagnosis and an offline system which runs at the end of each week, for novelty detection. All the X-rays collected during the week constitute the unlabeled set $U$ that the SSND method may then use for training. If a quickly spreading novel disease circulates the patient population, the detection model can identify the OOD samples that are then shown to the scarcely available experts. While the experts are examining the peculiar X-rays in the course of the next week, the model helps to collect more instances of the same new condition and can already encourage clinicians to practice extra caution when diagnosing these patients. Since the novelty detection algorithm is run every week, new diseases are identified with a delay of at most a week – the time it takes to collect an unlabeled set. Since new diseases emerge seldomly and the benefits of even delayed identification greatly outweigh the waiting time, SSND approaches are particularly suitable to this practical scenario.

**Summary and future work.** We would like to stress once again that a significant advantage of the SSND setting is that it does not require any labeled or oracle OOD data during fine-tuning, unlike many other related works summarized in Table 1. The only other approach that achieves comparable performance to our method (see Appendix E.2) uses a large transformer model pretrained on a much larger data set (Fort et al., 2021). At the same time, computationally, ERD reaches the optimal stopping time within the first 10 epochs on all the data sets we consider, which amounts to around 6 minutes of training time if the models in the ensemble are fine-tuned in parallel on NVIDIA 1080 Ti GPUs. Other ensemble diversification methods require training different models for each hyperparameter choice and have training losses that cannot be easily parallelized.

In conclusion, we propose a procedure that succeeds in exploiting unlabeled data to generate an ensemble with *regularized* disagreement, with remarkable novelty detection performance. We leave as future work a thorough investigation into the influence of the labeling scheme of the unlabeled set on the sample complexity of the method, as well as an analysis of the trade-off governed by the complexity of the model class of the classifiers.

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

# A    THEORETICAL STATEMENTS

**Definition A.1** (($\epsilon, \rho$)-clusterable data set). *We say that a data set $\mathcal{D} = \{(x_i, y_i)\}_{i=1}^n$ is ($\epsilon, \rho$)-clusterable for fixed $\epsilon > 0$ and $\rho \in [0, 1]$ if there exists a partitioning of it into subsets $\{C_1, ..., C_K\}$, which we call* clusters*, each with their associated unit-norm cluster center $c_i$, that satisfy the following conditions:*

- $\bigcup_{i=1}^K C_i = \mathcal{D}$ *and* $C_i \cap C_j = \emptyset, \forall i, j \in [K]$;

- *all the points in a cluster lie in the $\epsilon$-neighborhood of their corresponding cluster center, i.e. $||x - c_i||_2 \leq \epsilon$ for all $x \in C_i$ and all $i \in [K]$;*

- *a fraction of at least $1 - \rho$ of the points in each cluster $C_i$ have the same label, which we call the* cluster label *and denote $y^*(c_i)$. The remaining points suffer from label noise;*

- *if two cluster $C_i$ and $C_j$ have different labels, then their centers are $2\epsilon$ far from each other, i.e. $||c_i - c_j||_2 \geq 2\epsilon$;*

- *the clusters are balanced i.e. for all $i \in [K], \alpha_1 \frac{n}{K} \leq |C_i| \leq \alpha_2 \frac{n}{K}$, where $\alpha_1$ and $\alpha_2$ are two positive constants.*

In our case, for a fixed label $c \in \mathcal{Y}$, we assume that the set $S \cup (U, c)$ is ($\epsilon, \rho$)-clusterable into $K$ clusters. We further assume that each cluster $C_i$ only includes a few noisy samples from $(U_{\mathrm{ID}}^{\neg c}, c)$, i.e. $\frac{|C_i \cap (U_{\mathrm{ID}}^{\neg c}, c)|}{|C_i|} \leq \rho$ and that for clusters $C_i$ whose cluster label is not $c$, i.e. $y^*(c_i) \neq c$, it holds that $C_i \cap (U_{\mathrm{OOD}}, c) = \emptyset$.

We define the matrices $C := [c_1, ..., c_K]^T \in \mathbb{R}^{K \times d}$ and $\Sigma := (CC^T) \odot \mathbb{E}_g[\phi'(Cg)\phi'(Cg)^T]$, with $g \sim \mathcal{N}(0, I_d)$ and where $\odot$ denotes the elementwise product. We use $|| \cdot ||$ and $\lambda_{min}(\cdot)$ to denote the spectral norm and the smallest eigenvalue of a matrix, respectively.

For prediction, we consider a 2-layer neural network model with $p$ hidden units, where $p \gtrsim \frac{K^2 ||C||^4}{\lambda_{min}(\Sigma)^4}$. We can write this model as follows:

$$x \mapsto f(x; W) = v^T \phi(Wx), \tag{2}$$

The first layer weights $W$ are initialized with random values drawn from $\mathcal{N}(0, 1)$, while the last layer weights $v$ have fixed values: half of them are set to $1/p$ and the other half is $-1/p$. We consider activation functions $\phi$ with bounded first and second order derivatives, i.e. $|\phi'(x)| \leq \Gamma$ and $\phi''(x) \leq \Gamma$. We use the squared loss for training, i.e. $\mathcal{L}(W) = \frac{1}{2} \sum_{i=0}^n (y_i - f(x_i; W))^2$ and take gradient descent steps to find the optimum of the loss function, i.e. $W_{\tau+1} = W_\tau - \eta \nabla \mathcal{L}(W_\tau)$, where the step size is set to $\eta \simeq \frac{K}{n ||C||^2}$.

We can now state the following proposition:

**Proposition A.1.** *Assume that $\rho \leq \delta/8$ and $\epsilon \leq \alpha \delta \lambda_{min}(\Sigma)^2/K^2$, where $\delta$ is a constant such that $\delta \leq \frac{2}{|\mathcal{Y}-1|}$ and $\alpha$ is a constant that depends on $\overline{\Gamma}$. Then it holds with high probability $1 - 3/K^{100} - Ke^{-100d}$ over the initialization of the weights that the neural network trained on $S \cup (U, c)$ perfectly fits $S$, $(U_{\mathrm{ID}}^c, c)$ and $(U_{\mathrm{OOD}}, c)$, but not $(U_{\mathrm{ID}}^{\neg c}, c)$, after $T = c_4 \frac{||C||^2}{\lambda_{min}(\Sigma)}$ iterations.*

This result shows that there exists an optimal stopping time at which the neural network predicts the correct label on all ID points and the label $c$ on all the OOD points. As we will see later in the proof, the proposition is derived from a more general result which shows that the early stopped model predicts these labels not only on the points in $U$ but also in an $\epsilon$-neighborhood around cluster centers. Hence, an ERD ensemble can be used to detect holdout OOD samples similar to the ones in $U$, after being tuned on $U$. This follows the intuition that classifiers regularized with early stopping are smooth and generalize well.

The clusterable data model is generic enough to include data sets with non-linear decision boundaries. Moreover, notice that the condition in Proposition A.1 is satisfied when $S \cup (U_{\mathrm{ID}}, c)$ is ($\epsilon, \rho$)-clusterable and $(U_{\mathrm{OOD}}, c)$ is $\epsilon$-clusterable and if the cluster centers of $(U_{\mathrm{OOD}}, c)$ are at distance at least $2\epsilon$ from the cluster centers of $S \cup (U_{\mathrm{ID}}, c)$. A situation in which these requirements are met

is, for instance, when the OOD data comes from novel classes, when all classes (including the unseen ones that are not in the training set) are well separated, with cluster centers at least $2\epsilon$ away in Euclidean distance. In addition, in order to limit the amount of label noise in each cluster, it is necessary that the number of incorrectly labeled samples in $(U_{\text{ID}}^{\neg c}, c)$ is small, relative to the size of $S$.

In practice, we only need that the decision boundary separating $(U_{\text{OOD}}, c)$ from $S$ is easier to learn than the classifier required to interpolate the incorrectly labeled $(U_{\text{ID}}^{\neg c}, c)$, which is often the case, provided that $(U_{\text{OOD}}, c)$ is large enough and the OOD samples come from novel classes.

We now provide the proof for Proposition A.1:

*Proof.* We begin by restating a result from Li et al. (2020):

**Theorem A.1** (Li et al. (2020))**.** *Let $\mathcal{D} := \{(x_i, y_i)\} \in \mathbb{R}^d \times \mathcal{Y}$ be an $(\epsilon, \rho)$-clusterable training set, with $\epsilon \le c_1 \delta \lambda_{min}(\Sigma)^2 / K^2$ and $\rho \le \delta/8$, where $\delta$ is a constant that satisfies $\delta \le \frac{2}{|\mathcal{Y}|-1}$. Consider a two-layer neural network as described above, and train it with gradient descent starting from initial weights sampled i.i.d. from $\mathcal{N}(0, 1)$. Assume further that the step size is $\eta = c_2 \frac{K}{n \|C\|^2}$ and that the number of hidden units $p$ is at least $c_3 \frac{K^2 \|C\|^4}{\lambda_{min}(\Sigma)^4}$. Under these conditions, it holds with probability at least $1 - 3/K^{100} - Ke^{-100d}$ over the random draws of the initial weights, that after $T = c_4 \frac{\|C\|^2}{\lambda_{min}(\Sigma)}$ gradient descent steps, the neural network $x \mapsto f(x; W_T)$ predicts the correct cluster label for all points in the $\epsilon$-neighborhood of the cluster center, namely:*

$$\arg \max_{y \in \mathcal{Y}} |f(x; W_T) - \omega(y)| = y^*(c_i), \text{ for all } x \text{ with } \|x - c_i\|_2 \le \epsilon \text{ and all clusters } i \in [K], \quad (3)$$

*where $\omega : \mathcal{Y} \to \{0, 1\}^{|\mathcal{Y}|}$ yields one-hot embeddings of the labels. The constants $c_1, c_2, c_3, c_4$ depend only on $\Gamma$.*

Notice that, under the assumptions introduced above, the set $S \cup (U, c)$ is $(\epsilon, \rho)$-clusterable, since the incorrectly labeled ID points in $(U_{\text{ID}}^{\neg c}, c)$ constitute at most a fraction $\rho$ of the clusters they belong to. As a consequence, Proposition A.1 follows directly from Theorem A.1.

$\square$

# B DISAGREEMENT SCORE FOR OOD DETECTION

As we argue in Section 3.3, Algorithm 1 produces an ensemble that disagrees on OOD data, and hence, we want to devise a scalar score that reflects this model diversity. Previous works Lakshminarayanan et al. (2017); Ovadia et al. (2019) first average the softmax predictions of the models in the ensemble and then use the entropy as a metric, i.e. $(\text{H} \circ \text{Avg})(f_1(x), ..., f_K(x)) := -\sum_{i=1}^{|\mathcal{Y}|} (f(x))_i \log(f(x))_i$ where $f(x) := \frac{1}{K} \sum_{i=1}^{K} f_i(x)$ and $(f(x))_i$ is the $i^{\text{th}}$ element of $f(x) \in [0, 1]^{|\mathcal{Y}|}$[8]. We argue later that averaging discards information about the diversity of the models.

Recall that our average pairwise *disagreement* between the outputs of $K$ models in an ensemble reads:[9]

$$(\text{Avg} \circ \rho)(f_1(x), ..., f_K(x)) := \frac{2}{K(K-1)} \sum_{i \ne j} \rho(f_i(x), f_j(x)), \quad (4)$$

where $\rho$ is a measure of disagreement between the softmax outputs of two predictors, for example the total variation distance $\rho_{\text{TV}}(f_i(x), f_j(x)) = \frac{1}{2} \|f_i(x) - f_j(x)\|_1$ used in our experiments.

We briefly highlight the reason why averaging softmax outputs *first* like in previous works relinquishes all the benefits of having a more diverse ensemble, as opposed to the proposed pairwise

---

[8]We abuse notation slightly and denote our disagreement metric as $(\text{Avg} \circ \rho)$ to contrast it with the ensemble entropy metric $(\text{H} \circ \text{Avg})$, which first takes the average of the softmax outputs and only afterwards computes the score.

[9]We abuse notation slightly and denote our disagreement metric as $(\text{Avg} \circ \rho)$ to contrast it with the ensemble entropy metric $(\text{H} \circ \text{Avg})$, which first takes the average of the softmax outputs and only afterwards computes the score.

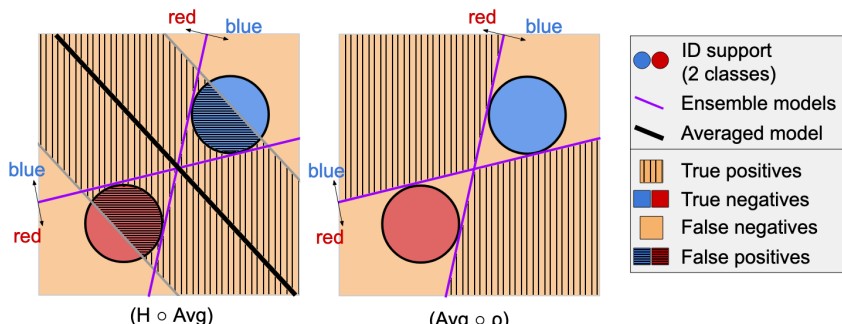

Figure 7: Cartoon illustration showing a diverse ensemble of linear binary classifiers. We compare OOD detection performance for two aggregation scores: (H ∘ Avg) (**Left**) and (Avg ∘ $\rho$) with $\rho(f_1(x), f_2(x)) = \mathbb{1}_{\text{sgn}(f_1(x)) \neq \text{sgn}(f_2(x))}$ (**Right**). The two metrics achieve similar TPRs, but using (H∘Avg) instead of our score, (Avg∘$\rho$), leads to more false positives, since the former simply flags as OOD a band around the averaged model (solid black line) and does not take advantage of the ensemble's diversity.

score in Equation 4. Recall that varying thresholds yield different true negative and true positive rates (TNR and TPR, respectively) for a given statistic. In the sketch in Figure 7 we show that the score we propose, (Avg ∘ $\rho$), achieves a higher TNR compared to (H ∘ Avg), for a fixed TPR, which is a common way of evaluating statistical tests. Notice that the detection region for (H ∘ Avg) is always limited to a band around the average model for any threshold value $t_0$. In order for the (H ∘ Avg) to have large TPR, this band needs to be wide, leading to many false positives. Instead, our disagreement score exploits the diversity of the models to more accurately detect OOD data.

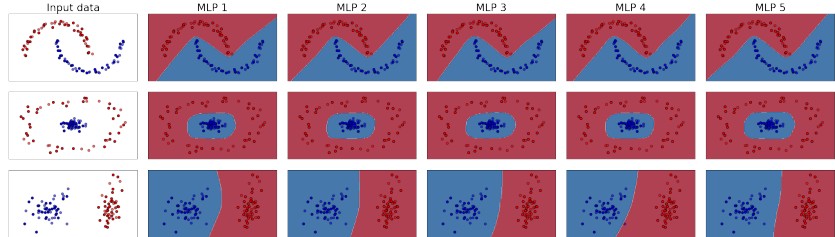

Figure 8: Relying only on the randomness of SGD and of the weight initialization to diversify models is not enough, as it often yields similar classifiers. Each column shows a different predictor trained from random initializations with Adam. All models have the same 1-hidden layer MLP architecture.

We now provide further quantitative evidence to support the intuition presented in Figure 7. The aggregation metric is tailored to exploit ensemble diversity, which makes it particularly beneficial for ERD. On the other hand, Vanilla Ensembles only rely on the stochasticity of the training process and the random initializations of the weights to produce diverse models, which often leads to classifiers that are strikingly similar as we show in Figure 8 for a few 2D data sets. As a consequence, using our disagreement score (Avg∘$\rho$) for Vanilla Ensembles can sometimes hurt OOD detection performance. To see this, consider the extreme situation in which the models in the ensemble are identical, i.e. $f_1 = f_2$. Then it follows that $(\text{Avg} \circ \rho)(f_1(x), f_2(x)) = 0$, for all test points $x$ and for any function $\rho$ that satisfies the distance axioms.

We note that the disagreement score that we propose takes a form that is similar to previous diversity scores, e.g. Zhang & Zhou (2010); Yu & Aizawa (2019). In the context of regression, one can measure uncertainty using the variance of the outputs metric previously employed in works such as Gal & Ghahramani (2016). However, we point out that using the output variance requires that the ensemble is the result of sampling from a random process (e.g. sampling different training data for the models, or sampling different parameters from a posterior). In our framework, we obtain the ensemble by solving a different optimization problem for each of the models by assigning a different label to the unlabeled data. Therefore, despite their similarities, our disagreement score and the output variance are, on a conceptual level, fundamentally different metrics.

Table 3 shows that $(\text{Avg} \circ \rho)$ leads to worse OOD detection performance for Vanilla Ensembles, compared to using the entropy of the average softmax score, $(\text{H} \circ \text{Avg})$, which was proposed in prior work. However, if the ensembles are indeed diverse, as we argue is the case for our method ERD (see Section 3.3), then there is a clear advantage to using a score that, unlike $(\text{H} \circ \text{Avg})$, takes diversity into account, as shown in Table 3.

Table 3: The disagreement score that we propose $(\text{Avg} \circ \rho)$ exploits ensemble diversity and benefits in particular ERD ensembles. OOD detection performance is significantly improved when using $(\text{Avg} \circ \rho)$ compared to the previously proposed $(\text{H} \circ \text{Avg})$ metric. Since Vanilla Ensemble are not diverse enough, a score that relies on model diversity can hurt OOD detection performance. We highlight the AUROC and the TNR@95 obtained with the score function that is ***best for Vanilla Ensemble*** and the **best for ERD**.

| ID data | OOD data | Vanilla Ensembles $(\text{H} \circ \text{Avg})$ | Vanilla Ensembles $(\text{Avg} \circ \rho)$ | ERD $(\text{H} \circ \text{Avg})$ | ERD $(\text{Avg} \circ \rho)$ |
|---|---|---|---|---|---|
| | | | AUROC ↑ / TNR@95 ↑ | | |
| SVHN | CIFAR10 | *0.97 / 0.88* | 0.96 / *0.89* | 0.86 / 0.85 | **0.99 / 0.97** |
| CIFAR10 | SVHN | *0.92 / 0.78* | 0.91 / *0.78* | 0.92 / 0.92 | **1.00 / 1.00** |
| CIFAR100 | SVHN | *0.84 / 0.48* | 0.79 / 0.46 | 0.36 / 0.35 | **1.00 / 1.00** |
| SVHN[0:4] | SVHN[5:9] | *0.92 / 0.69* | 0.91 / *0.69* | **0.94 / 0.66** | **0.94 / 0.66** |
| CIFAR10[0:4] | CIFAR10[5:9] | *0.80 / 0.39* | *0.80 / 0.39* | **0.91** / 0.65 | **0.91 / 0.66** |
| CIFAR100[0:49] | CIFAR100[50:99] | *0.78 / 0.35* | 0.76 / 0.34 | 0.63 / 0.38 | **0.81 / 0.40** |
| Average | | *0.87 / 0.60* | 0.86 / 0.59 | 0.77 / 0.64 | **0.94 / 0.78** |

We highlight once again that other methods that attempt to obtain diverse ensembles, such as MCD, fail to train models with sufficient disagreement, even when they use oracle OOD for hyperparameter tuning (Figure 9a).

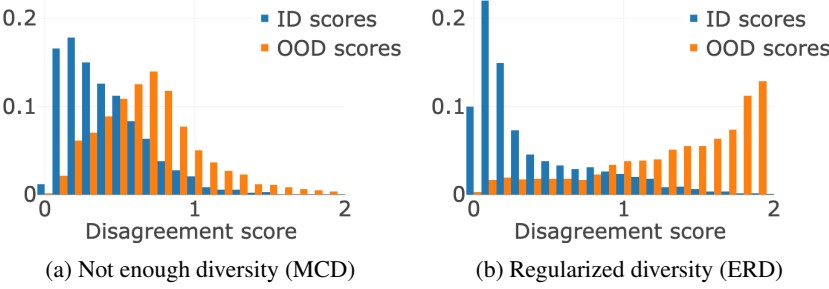

(a) Not enough diversity (MCD)          (b) Regularized diversity (ERD)

Figure 9: Distribution of disagreement scores on ID and OOD data for an ensemble that is not diverse enough (**Left**), and an ensemble with regularized disagreement (**Right**). Note that MCD is early-stopped using oracle OOD data. ID=CIFAR10[0:4], OOD=CIFAR10[5:9].

## C  EXPERIMENT DETAILS

### C.1  BASELINES

In this section we describe in detail the baselines with which we compare our method and describe how we choose their hyperparameters. For all baselines we use the hyperparameters suggested by the authors for the respective data sets (e.g. different hyperparameters for CIFAR10 or ImageNet). For all methods, we use pretrained models provided by the authors. However, we note that for the novel-class settings, pretraining on the entire training set means that the model is exposed to the OOD classes as well, which is undesirable. Therefore, for these settings we pretrain only on the split of the training set that contains the ID classes. Since the classification problem is similar to the original one of training on the entire training set, we use the same hyperparameters that the authors report in the original papers.

Moreover, we point out that even though different methods use different model architectures, that is not inherently unreasonable when the goal is OOD detection, since it is not clear if a complex model is more desirable than a smaller model. For this reason, we use the model architecture recommended by the authors of the baselines and which was used to produce the good results reported in their

published works. For Vanilla Ensembles and for ERD we show results for different architectures in Appendix E.7.

- **Vanilla Ensembles** Lakshminarayanan et al. (2017): We train an ensemble on the training set according to the true labels. For a test sample, we average the outputs of the softmax probabilities predicted by the models, and use the entropy of the resulting distribution as the score for the hypothesis test described in Section 3.2. We use ensembles of 5 models, with the same architecture and hyperparameters as the ones used for ERD. Hyperparameters are tuned to achieve good validation accuracy.

- **Gram method** Sastry & Oore (2019): The Gram baseline is similar to the Mahalanobis method in that both use the intermediate feature representations obtained with a deep neural network to determine whether a test point is an outlier. However, what sets the Gram method apart is the fact that it does not need any OOD data for training or calibration. We use the pretrained models provided by the authors, or train our own, using the same methodology as described for the Mahalanobis baseline. For OOD detection, we use the code published by the authors. We note that for MLP models, the Gram method is difficult to tune and we could not find a configuration that works well, despite our best efforts and following the suggestions proposed during our communication with the authors.

- **Deep Prior Networks (DPN)** Malinin & Gales (2018): DPN is a Bayesian Method that trains a neural network (Prior Network) to parametrize a Dirichlet distribution over the class probabilities. We train a WideResNet WRN-28-10 for $100$ epochs using SGD with momentum $0.9$, with an initial learning rate of $0.01$, which is decayed by $0.2$ at epochs $50$, $70$, and $90$. For MNIST, we use EMINST/Letters as OOD for tuning. For all other settings, we use TinyImages as OOD for tuning.

- **Outlier Exposure** Hendrycks et al. (2019): This approach makes a model's softmax predictions close to the uniform distribution on the known outliers, while maintaining a good classification performance on the training distribution. We use the WideResNet architecture (WRN) Zagoruyko & Komodakis (2016). For fine-tuning, we use the settings recommended by the authors, namely we train for $10$ epochs with learning rate $0.001$. For training from scratch, we train for $100$ epochs with an initial learning rate of $0.1$. When the training data set is either CIFAR10/CIFAR100 or ImageNet, we use the default WRN parameters of the author's code, namely $40$ layers, $2$ widen-factor, droprate $0.3$. When the training dataset is SVHN, we use the author's recommended parameters of $16$ layers, $4$ widen-factor and droprate $0.4$. All settings use the cosine annealing learning rate scheduler provided with the author's code, without any modifications. For all settings, we use TinyImages as known OOD data during training. In Section E.5 we show results for known OOD data that is similar to the OOD data used for testing.

- **Mahalanobis** Lee et al. (2018): The method pretrains models on the labeled training data. For a test data point, it uses the intermediate representations of each layer as "extracted features". It then performs binary classification using logistic regression using these extracted features. In the original setting, the classification is done on "training" ID vs "training" OOD samples (which are from the same distribution as the test OOD samples). Furthermore, hyperparameter tuning for the optimal amount of noise is performed on validation ID and OOD data. We use the WRN-28-10 architecture, pretrained for $200$ epochs. The initial learning rate is $0.1$, which is decayed at epochs $60$, $120$, and $160$ by $0.2$. We use SGD with momentum $0.9$, and the standard weight decay of $5 \cdot 10^{-4}$. The code published for the Mahalanobis method performs a hyperparameter search automatically for each of the data sets.

The following baselines assume the same *Unknown OOD* setting as ERD, in which one has access to both a labeled ID training set $S$ and an unlabeled set with an unknown mixture of ID and OOD samples $U$.

- **Non-negative PU learning (nnPU)** Kiryo et al. (2017): The method trains a binary predictor to distinguish between a set of known positives (in our case the ID data) and a set that contains a mixture of positives and negatives (in our case the unlabeled set). To prevent the interpolation of all the unlabeled samples, Kiryo et al. (2017) proposes a regularized objective. It is important to note that most training objectives in the PU learning literature

require that the ratio between the positives and negatives in the unlabeled set is known or easy to estimate. For our experiments we always use the exact OOD ratio to train the nnPU baseline. Therefore, we obtain an upper bound on the AUROC/TNR@95. If the ratio is estimated from finite samples, then estimation errors may lead to slightly worse OOD detection performance. We perform a grid search over the learning rate and the threshold that appears in the nnPU regularizer and pick the option with the best validation accuracy measured on a holdout set with only positive samples (in our case, ID data).

- **Maximum Classifier Discrepancy (MCD)** Yu & Aizawa (2019): The MCD method trains two classifiers at the same time, and makes them disagree on the unlabeled data, while maintaining good classification performance. We use the WRN-28-10 architecture as suggested in the paper. We did not change the default parameters which came with the author's code, so weight decay is $10^{-4}$, and the optimizer is SGD with momentum $0.9$. When available (for CIFAR10 and CIFAR100), we use the pretrained models provided by the authors. For the other training datasets, we use their methodology to generate pretrained models: We train a WRN-28-10 for 200 epochs. The learning rate starts at 0.1 and drops by a factor of 10 at $50\%$ and $75\%$ of the training progress.

- **Mahalanobis-U**: This is a slightly different version of the Mahalanobis baseline, for which we use early-stopped logistic regression to distinguish between the training set and an unlabeled set with ID and OOD samples (instead of discriminating a known OOD set from the inliers). The early stopping iteration is chosen to minimize the classification errors on a validation set that contains only ID data (recall that we do not assume to know which are the OOD samples).

In addition to these approaches that have been introduced in prior work, we also propose a strong novel baseline that that bares some similarity to PU learning and to ERD.

- **Binary classifier** The approach consists in discriminating between the labeled ID training set and the mixed unlabeled set, that contains both ID and OOD data. We use regularization to prevent the trivial solution for which the entire unlabeled set is predicted as OOD. Unlike PU learning, the binary classifier does not require that the OOD ratio in the test distribution is known. The approach is similar to a method described in Scott & Blanchard (2008) which also requires that the OOD ratio of the unlabeled set is known. We tune the learning rate and the weight of the unlabeled samples in the training loss by performing a grid search and selecting the configuration with the best validation accuracy, computed on a holdout set containing only ID samples. We note that the binary classifier that appears in Section F in the medical benchmark, is not the same as this baseline. For more details on the binary classifier that appears in the medical data experiments we refer the reader to Cao et al. (2020).

## C.2 Training configuration for ERD

For ERD we always use hyperparameters that give the best validation accuracy when training a model on the ID training set. In other words, we pick hyperparameter values that lead to good ID generalization and do not perform further hyperparameter tuning for the different OOD data sets on which we evaluate our approach. We point out that, if the ID labeled set is known to suffer from class imbalance, subpopulation imbalance or label noise, any training method that addresses these issues can be used instead of standard empirical risk minimization to train our ensemble (e.g. see the works of Sagawa et al. (2020); Li et al. (2020) etc).

For MNIST and FashionMNIST, we train ensembles of 3-layer MLP models with ReLU activations. Each intermediate layer has 100 neurons. The models are optimized using Adam, with a learning rate of $0.001$, for 10 epochs.

For SVHN, CIFAR10/CIFAR100 and ImageNet, we train ensembles of ResNet20 He et al. (2016). The models are initialized with weights pretrained for $100$ epochs on the labeled training set. We fine-tune each model for 10 epochs using SGD with momentum $0.9$, and a learning rate of $0.001$. The weights are trained with an $\ell_2$ regularization coefficient of $5e-4$. We use a batch size of 128 for all scenarios, unless explicitly stated otherwise. We used the same hyperparameters for all settings.

For pretraining, we perform SGD for 100 epochs and use the same architecture and hyperparameters as described above, with the exception of the learning rate that starts at $0.1$, and is multiplied by $0.2$ at epochs $50$, $70$ and $90$.

Apart from ERD, which fine-tunes the ensemble models starting from pretrained weights, we also present in the Appendix results for ERD++. This variant of our method trains the models from random initializations, and hence needs more iterations to converge, making it more computationally expensive than ERD. We train all models in the ERD++ ensembles for 100 epochs with a learning rate that starts at $0.1$, and is multiplied by $0.2$ at epochs $50$, $70$ and $90$. All other hyperparameters are the same as for ERD ensembles.

For the medical data sets, we train a Densenet-121 as the authors do in the original paper Cao et al. (2020). For ERD++, we do not use random weight initializations, but instead we start with the ImageNet weights provided with Tensorflow. The training configuration is exactly the same as for ResNet20, except that we use a batch size of 32 due to GPU memory restrictions, and for fine tuning we use a constant learning rate of $10^{-5}$.

## D   ID AND OOD DATA SETS

### D.1   DATA SETS

For evaluation, we use the following image data sets: MNIST Lecun et al. (1998), Fashion MNIST Xiao et al. (2017), SVHN Netzer et al. (2011), CIFAR10 and CIFAR100 Krizhevsky (2009).

For the experiments using MNIST and FashionMNIST the training set size is 50K, the validation size is 10K, and the test ID and test OOD sizes are both 10K. For SVHN, CIFAR10 and CIFAR100, the training set size is 40K, the validation size is 10K, and the unlabeled set contains 10K samples: 5K are ID and 5K are OOD. For evaluation, we use a holdout set of 10K examples (half ID, half OOD). For the settings that use half of the classes as ID and the other half as OOD, all the sizes are divided by 2.

### D.2   SAMPLES FOR THE SETTINGS WITH NOVEL CLASSES

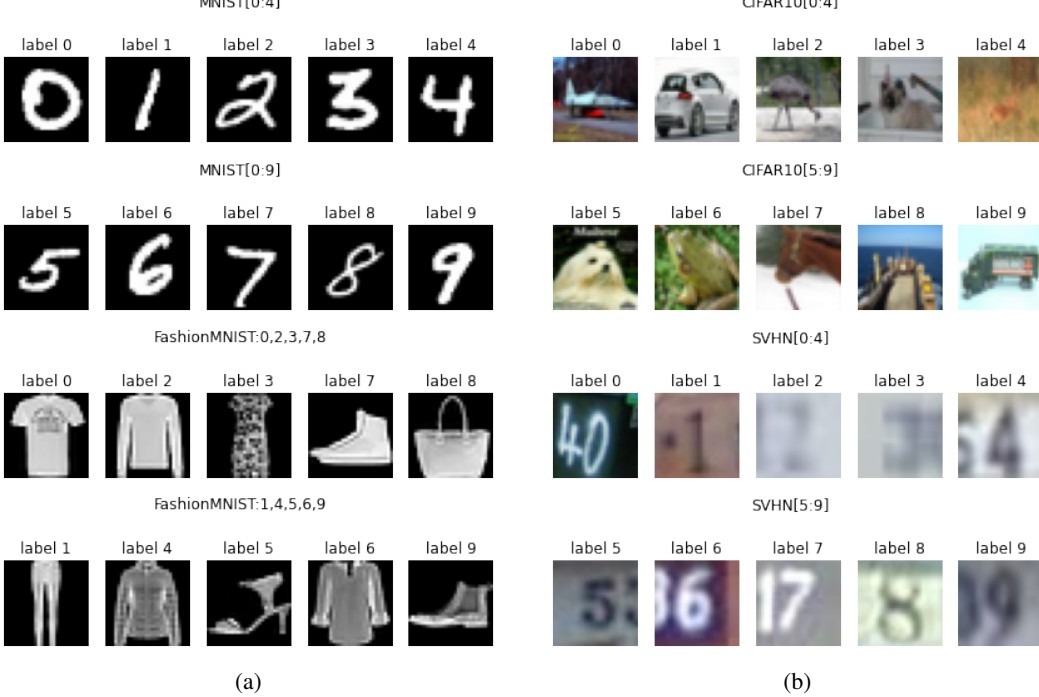

Figure 10: (a) Data samples for the MNIST/FashionMNIST splits. (b) Data samples for the CIFAR10/SVHN splits.

# E    MORE EXPERIMENTS

## E.1    EVALUATION ON THE UNLABELED SET

In the main text we describe how one can leverage the unlabeled set $U$ to obtain an OOD detection algorithm that accurately identifies outliers at test time that similar to the ones in $U$. It is, however, possible to also use our method ERD to flag the OOD samples contained in the same set $U$ used for fine-tuning the ensemble. In Table 4 we show that the OOD detection performance of ERD is similar regardless of whether we use $U$ for evaluation, or a holdout test set $T$ drawn from the same distribution as $U$.

Table 4: Comparison between the OOD detection performance of ERD when using a holdout test set $T$ for evaluation, or the same unlabeled set $U$ that was used for fine-tuning the models.

| ID data | OOD data | ERD (eval on $T$) AUROC ↑ / TNR@95 ↑ | ERD (eval on $U$) |
|---------|----------|------------------|------------------|
| SVHN | CIFAR10 | 1.00 / 0.99 | 1.00 / 0.99 |
| CIFAR10 | SVHN | 1.00 / 1.00 | 1.00 / 1.00 |
| CIFAR100 | SVHN | 1.00 / 1.00 | 1.00 / 1.00 |
| FMNIST[0,2,3,7,8] | FMNIST[1,4,5,6,9] | 0.94 / 0.67 | 0.94 / 0.67 |
| SVHN[0:4] | SVHN[5:9] | 0.95 / 0.74 | 0.96 / 0.79 |
| CIFAR10[0:4] | CIFAR10[5:9] | 0.93 / 0.70 | 0.93 / 0.69 |
| CIFAR100[0:49] | CIFAR100[50:99] | 0.82 / 0.44 | 0.80 / 0.36 |
| Average | | 0.95 / 0.79 | 0.95 / 0.79 |

## E.2    COMPARISON WITH OTHER RELATED WORKS

We compare our method to more OOD detection approaches. For various reasons we did not run these methods ourselves on the data sets for which we evaluate our method in Section 4 (e.g. resource constraints, code not available, unable to replicate published results, poor performance reported by the authors etc). We collected the AUROC numbers presented in Table 5 from the papers that introduce each method. We note that our approach shows an excellent overall performance, almost matching the AUROC of the best performing method of Fort et al. (2021) which uses large scale visual transformer models pretrained on a superset of the OOD data, i.e. ImageNet21k.

Furthermore, we note that generative models (Nalisnick et al., 2019; Choi et al., 2018; Akçay et al., 2018) and one-class classification approaches (Ruff et al., 2020; Tack et al., 2020; Sohn et al., 2021) showed generally bad performance, in particular on near OOD data. When the ID training set is made up of several diverse classes, it is difficult to represent accurately all the ID data, and only the ID data.

Table 5: AUROC numbers collected from the literature for a number of relevant OOD detection methods. We note that the method of Fort et al. (2021) ([†]) uses a large scale visual transformer models pretrained on a superset of the OOD data, i.e. ImageNet21k, while the method of Sehwag et al. (2021) ([*]) uses oracle OOD samples for training from the same data set as test OOD. For the settings with random classes, the numbers are averages over 5 draws and the standard deviation is always strictly smaller than 0.01 for our method.

| ID data | OOD data | Fort et al. (2021)[†] | Zhang et al. (2020a) | Winkens et al. (2020) | Tack et al. (2020) | Sehwag et al. (2021)[*] | Liu & Abbeel (2020) | Zhang et al. (2020b) | ERD (ours) | ERD++ (ours) |
|---------|----------|------|------|------|------|------|------|------|------|------|
| CIFAR10 | CIFAR100 | 98.52 | 0.95 | 0.92 | 0.92 | 0.93 | 0.91 | - | 0.92 | 0.95 |
| CIFAR100 | CIFAR10 | 96.23 | 0.85 | 0.78 | - | 0.78 | - | - | 0.91 | 0.94 |
| SVHN: 6 random classes | SVHN: 4 random classes | - | 0.94 | - | - | - | - | 0.91 | 0.94 | 0.94 |
| CIFAR10: 6 random classes | CIFAR10: 4 random classes | - | 0.94 | - | - | - | - | 0.85 | 0.94 | 0.97 |

## E.3    OOD DETECTION FOR DATA WITH COVARIATE SHIFT

In this section we evaluate the baselines and the method that we propose on settings in which the OOD data suffers from covariate shift Shimodaira (2000). The goal is to identify all samples that

come from the shifted distribution, regardless of how strong the shift is. Notice that mild shifts may be easier to tackle by domain adaptation algorithms, but when the goal is OOD detection they pose a much more difficult challenge.

We want to stress that in practice one may not be interested in identifying *all* samples with distribution shift as OOD, since a classifier may still produce correct predictions on some of them. In contrast, when data suffers from covariate shift we can try to learn predictors that perform well on both the training and the test distribution, and we may use a measure of predictive uncertainty to identify only those test samples on which the classifier cannot make confident predictions. Nevertheless, we use these covariate shift settings as a challenging OOD detection benchmark and show in Table 7 that our method ERD does indeed outperform prior baselines on these difficult settings.

We use as outliers corrupted variants of CIFAR10 and CIFAR100 Hendrycks & Dietterich (2019), as well as a scenario where ImageNet Deng et al. (2009) is used as ID data and ObjectNet Barbu et al. (2019) as OOD, both resized to 32x32. Figure 11 shows samples from these data sets. The Gram and nnPU baselines do not give satisfactory results on the difficult CIFAR10/CIFAR100 settings in Table 2 and thus we do not consider them for the distribution shift cases. For the *Unknown OOD* methods (i.e. MCD, Mahal-U and ERD/ERD++) we evaluate on the same unlabeled set that is used for training (see the discussion in Section E.1).

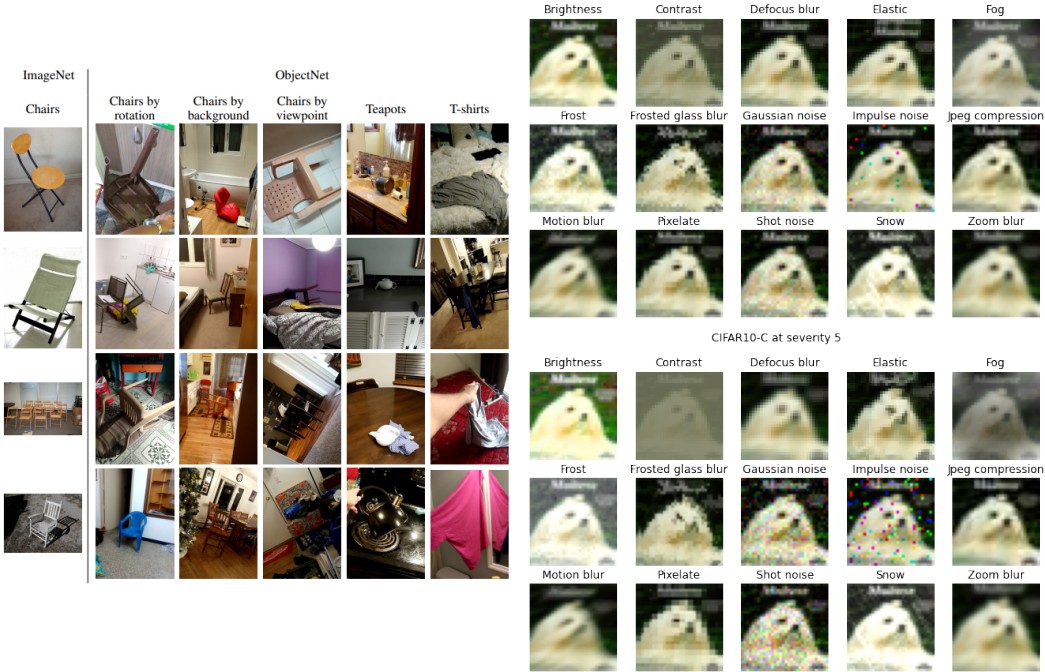

Figure 11: Left: Samples from ImageNet and ObjectNet taken from the original paper by Barbu et al. (2019). Right: Data samples for the corrupted CIFAR10-C data set.

Furthermore, we present results on distinguishing between CIFAR10 Krizhevsky (2009) and CIFAR10v2 Recht et al. (2018), a data set meant to be drawn from the same distribution as CIFAR10 (generated from the Tiny Images collection Torralba et al. (2008)). In Recht et al. (2019), the authors argue that CIFAR10 and CIFAR10v2 come from very similar distributions. They provide supporting evidence by training a binary classifier to distinguish between them, and observing that the accuracy that is obtained of 52.9% is very close to random.

Our experiments show that the two data sets are actually distinguishable, contrary to what previous work has argued. First, our own binary classifier trained on CIFAR10 vs CIFAR10v2 obtains a test accuracy of 67%, without any hyperparameter tuning. The model we use is a ResNet20 trained for 200 epochs using SGD with momentum 0.9. The learning rate is decayed by 0.2 at epochs 90, 140, 160 and 180. We use 1600 examples from each data set for training, and we validate using 400 examples from each data set.

Table 6: OOD detection performance on CIFAR10 vs CIFAR10v2

| ID data | OOD data | Vanilla Ensembles | DPN | OE | Mahal. | MCD | Mahal-U | ERD | ERD++ |
|---|---|---|---|---|---|---|---|---|---|
| | | | | | AUROC ↑ / TNR@95 ↑ | | | | |
| CIFAR10 | CIFAR10v2 | *0.64* / *0.13* | 0.63 / 0.09 | *0.64* / 0.12 | 0.55 / 0.08 | 0.58 / 0.10 | 0.56 / 0.07 | 0.76 / 0.26 | **0.91** / **0.80** |

Our OOD detection experiments (presented in Table 6) show that most baselines are able to distinguish between the two data sets, with ERD achieving the highest performance. The methods which require OOD data for tuning (Outlier Exposure and DPN) use CIFAR100.

Table 7: OOD detection performance on data with covariate shift. For ERD and vanilla ensembles, we train 5 ResNet20 models for each setting. The evaluation metrics are computed on the unlabeled set.

| ID data | OOD data | Vanilla Ensembles | DPN | OE | Mahal. | MCD | Mahal-U | ERD | ERD++ |
|---|---|---|---|---|---|---|---|---|---|
| | | | | | AUROC ↑ / TNR@95 ↑ | | | | |
| CIFAR10 | CIFAR10-C sev 2 (A) | 0.68 / 0.20 | 0.73 / 0.31 | 0.70 / 0.20 | *0.84* / *0.53* | 0.82 / 0.50 | 0.75 / 0.38 | 0.96 / 0.86 | **0.99** / **0.95** |
| CIFAR10 | CIFAR10-C sev 2 (W) | 0.51 / 0.05 | 0.47 / 0.03 | 0.52 / 0.06 | *0.58* / *0.08* | 0.52 / 0.06 | 0.55 / 0.07 | 0.68 / 0.19 | **0.86** / **0.41** |
| CIFAR10 | CIFAR10-C sev 5 (A) | 0.84 / 0.49 | 0.89 / 0.60 | 0.86 / 0.54 | 0.94 / 0.80 | *0.95* / *0.84* | 0.88 / 0.63 | **1.00** / 0.99 | **1.00** / **1.00** |
| CIFAR10 | CIFAR10-C sev 5 (W) | 0.60 / 0.10 | 0.72 / 0.10 | 0.63 / 0.11 | *0.78* / *0.27* | 0.60 / 0.08 | 0.68 / 0.12 | 0.98 / 0.86 | **1.00** / **1.00** |
| CIFAR100 | CIFAR100-C sev 2 (A) | 0.68 / 0.20 | 0.62 / 0.18 | 0.65 / 0.19 | *0.82* / *0.48* | 0.72 / 0.29 | 0.67 / 0.22 | 0.94 / 0.76 | **0.97** / **0.86** |
| CIFAR100 | CIFAR100-C sev 2 (W) | 0.52 / 0.06 | 0.32 / 0.03 | 0.52 / 0.06 | *0.55* / *0.07* | 0.52 / 0.06 | 0.55 / 0.06 | 0.71 / 0.19 | **0.86** / **0.44** |
| CIFAR100 | CIFAR100-C sev 5 (A) | 0.78 / 0.37 | 0.74 / 0.36 | 0.76 / 0.37 | *0.92* / *0.72* | 0.91 / 0.65 | 0.84 / 0.55 | 0.99 / 0.97 | **1.00** / **0.99** |
| CIFAR100 | CIFAR100-C sev 5 (W) | 0.64 / 0.14 | 0.49 / 0.12 | 0.62 / 0.13 | *0.71* / *0.19* | 0.60 / 0.10 | 0.63 / 0.13 | 0.96 / 0.71 | **0.98** / **0.89** |
| Tiny ImageNet | Tiny ObjectNet | 0.82 / 0.49 | 0.70 / 0.32 | 0.79 / 0.37 | 0.75 / 0.26 | *0.99* / *0.98* | 0.72 / 0.25 | 0.98 / 0.88 | **0.99** / **0.98** |
| | Average | 0.67 / 0.23 | 0.63 / 0.23 | 0.67 / 0.23 | *0.76* / 0.38 | 0.74 / *0.39* | 0.70 / 0.27 | 0.91 / 0.71 | **0.96** / **0.83** |

### E.4 RESULTS WITH A SMALLER UNLABELED SET

We now show that our method performs well even when the unlabeled set is significantly smaller. In particular, we show in the table below that ERD maintains a high AUROC and TNR@95 even when only 1,000 unlabeled samples are used for fine-tuning (500 ID and 500 OOD).

Table 8: Experiments with a test set of size 1,000, with an equal number of ID and OOD test samples. For ERD and vanilla ensembles, we train 5 ResNet20 models for each setting. The evaluation metrics are computed on the unlabeled set.

| ID data | OOD data | Vanilla Ensembles | DPN | OE | Mahal. | MCD | Mahal-U | ERD |
|---|---|---|---|---|---|---|---|---|
| | | | | | AUROC ↑ / TNR@95 ↑ | | | |
| SVHN | CIFAR10 | 0.97 / 0.88 | *1.00* / *1.00* | *1.00* / *1.00* | 0.99 / 0.98 | 0.97 / 0.85 | 0.99 / 0.95 | **1.00** / **0.99** |
| CIFAR10 | SVHN | 0.92 / 0.78 | 0.95 / 0.85 | 0.97 / 0.89 | *0.99* / *0.96* | 1.00 / 0.98 | 0.99 / 0.96 | **1.00** / **1.00** |
| CIFAR100 | SVHN | 0.84 / 0.48 | 0.77 / 0.44 | 0.82 / 0.50 | *0.98* / *0.90* | 0.97 / 0.73 | 0.98 / 0.92 | **0.99** / **1.00** |
| SVHN[0:4] | SVHN[5:9] | *0.92* / 0.69 | 0.87 / 0.19 | 0.85 / 0.52 | *0.92* / *0.71* | 0.91 / 0.51 | 0.91 / 0.63 | **0.97** / **0.86** |
| CIFAR10[0:4] | CIFAR10[5:9] | 0.80 / 0.39 | *0.82* / 0.32 | *0.82* / *0.41* | 0.79 / 0.27 | 0.69 / 0.25 | 0.64 / 0.13 | **0.87** / **0.50** |
| CIFAR100[0:49] | CIFAR100[50:99] | *0.78* / *0.35* | 0.70 / 0.26 | 0.74 / 0.31 | 0.72 / 0.20 | 0.70 / 0.26 | 0.72 / 0.19 | **0.79** / **0.38** |
| CIFAR10 | CIFAR10-C sev 2 (A) | 0.68 / 0.20 | 0.73 / 0.31 | 0.70 / 0.20 | *0.84* / *0.53* | 0.82 / 0.50 | 0.75 / 0.38 | **0.91** / **0.71** |
| CIFAR10 | CIFAR10-C sev 2 (W) | 0.51 / 0.05 | 0.47 / 0.03 | 0.52 / 0.06 | *0.58* / *0.08* | 0.52 / 0.06 | 0.55 / 0.07 | **0.57** / **0.09** |
| CIFAR10 | CIFAR10-C sev 5 (A) | 0.84 / 0.49 | 0.89 / 0.60 | 0.86 / 0.54 | *0.94* / *0.80* | 0.95 / 0.84 | 0.88 / 0.63 | **0.99** / **0.95** |
| CIFAR10 | CIFAR10-C sev 5 (W) | 0.60 / 0.10 | 0.72 / 0.10 | 0.63 / 0.11 | *0.78* / *0.27* | 0.60 / 0.08 | 0.68 / 0.12 | **0.92** / **0.67** |
| CIFAR100 | CIFAR100-C sev 2 (A) | 0.68 / 0.20 | 0.62 / 0.18 | 0.65 / 0.19 | *0.82* / *0.48* | 0.72 / 0.29 | 0.67 / 0.22 | **0.84** / **0.48** |
| CIFAR100 | CIFAR100-C sev 2 (W) | 0.52 / 0.06 | 0.32 / 0.03 | 0.52 / 0.06 | *0.55* / *0.07* | 0.52 / 0.06 | **0.55** / 0.06 | **0.55** / **0.07** |
| CIFAR100 | CIFAR100-C sev 5 (A) | 0.78 / 0.37 | 0.74 / 0.36 | 0.76 / 0.37 | *0.92* / *0.72* | 0.91 / 0.65 | 0.84 / 0.55 | **0.96** / **0.80** |
| CIFAR100 | CIFAR100-C sev 5 (W) | 0.64 / 0.14 | 0.49 / 0.12 | 0.62 / 0.13 | *0.71* / *0.19* | 0.60 / 0.10 | 0.63 / 0.13 | **0.81** / **0.25** |
| | Average | 0.75 / 0.37 | 0.72 / 0.34 | 0.75 / 0.38 | *0.82* / *0.51* | 0.78 / 0.44 | 0.77 / 0.42 | **0.87** / **0.62** |

### E.5    More results for Outlier Exposure

Table 9: Results for Outlier Exposure, when using the same corruption type, but with a higher/lower severity, as OOD data seen during training.

| ID data | OOD data | OE (trained on sev5) | OE (trained on sev2) |
|---------|----------|---------------------|---------------------|
| | | AUROC ↑ | |
| CIFAR10 | CIFAR10-C sev 2 (A) | 0.89 | N/A |
| CIFAR10 | CIFAR10-C sev 2 (W) | 0.65 | N/A |
| CIFAR10 | CIFAR10-C sev 5 (A) | N/A | 0.98 |
| CIFAR10 | CIFAR10-C sev 5 (W) | N/A | 0.78 |
| CIFAR100 | CIFAR100-C sev 2 (A) | 0.85 | N/A |
| CIFAR100 | CIFAR100-C sev 2 (W) | 0.59 | N/A |
| CIFAR100 | CIFAR100-C sev 5 (A) | N/A | 0.97 |
| CIFAR100 | CIFAR100-C sev 5 (W) | N/A | 0.67 |
| | Average | 0.87 | 0.98 |

The Outlier Exposure method needs access to a set of OOD samples during training. The numbers we report in the rest of paper for Outlier Exposure are obtained by using the TinyImages data set as the OOD samples that are seen during training. In this section we explore the use of an $OOD_{train}$ data set that is more similar to the OOD data observed at test time. This is a much easier setting for the Outlier Exposure method: the closer $OOD_{train}$ is to $OOD_{test}$, the easier it will be for the model tuned on $OOD_{train}$ to detect the test OOD samples.

In Table 9 we focus only on the settings with corruptions. For each corruption type, we use the lower severity corruption as $OOD_{train}$ and evaluate on the higher severity data and vice versa. We report for each metric the average taken over all corruptions (A), and the value for the worst-case setting (W).

### E.6    Results on MNIST and FashionMNIST

Table 10: Results on MNIST/FashionMNIST settings. For ERD and vanilla ensembles, we train 5 3-hidden layer MLP models for each setting. The evaluation metrics are computed on the unlabeled set.

| ID data | OOD data | Vanilla Ensembles | DPN | OE | Mahal. | nnPU | MCD | Mahal-U | Bin. Classif. | ERD | ERD++ |
|---------|----------|-------------------|-----|-----|--------|------|-----|---------|---------------|-----|-------|
| | | | | | | AUROC ↑ / TNR@95 ↑ | | | | | |
| MNIST | FMNIST | 0.81 / 0.01 | *1.00 / 1.00* | *1.00 / 1.00* | *1.00 / 1.00* | *1.00 / 1.00* | *1.00 / 1.00* | 1.00 / 0.98 | *1.00 / 1.00* | 1.00 / 1.00 | __1.00__ / __1.00__ | __1.00__ / __1.00__ |
| FMNIST | MNIST | 0.87 / 0.42 | *1.00 / 1.00* | 0.68 / 0.16 | 0.99 / 0.97 | *1.00 / 1.00* | *1.00 / 1.00* | 0.99 / 0.96 | 1.00 / 1.00 | __1.00__ / __1.00__ | __1.00__ / __1.00__ |
| MNIST[0:4] | MNIST[5:9] | 0.94 / 0.72 | *0.99* / 0.97 | 0.95 / 0.78 | *0.99 / 0.98* | 0.99 / 0.97 | 0.96 / 0.76 | *0.99 / 0.98* | 0.99 / 0.94 | __0.99__ / 0.96 | __0.99__ / __0.97__ |
| FMNIST[0,2,3,7,8] | FMNIST[1,4,5,6,9] | 0.64 / 0.07 | 0.77 / 0.15 | 0.66 / 0.12 | 0.77 / 0.20 | *0.95 / 0.71* | 0.78 / 0.30 | 0.82 / 0.39 | 0.95 / 0.66 | __0.94__ / 0.67 | __0.94__ / __0.68__ |
| | Average | 0.82 / 0.30 | 0.94 / 0.78 | 0.82 / 0.51 | 0.94 / 0.79 | *0.98 / 0.92* | 0.94 / 0.76 | 0.95 / 0.83 | 0.98 / 0.90 | __0.98__ / __0.91__ | __0.98__ / __0.91__ |

For FashionMNIST we chose this particular split (i.e. classes 0,2,3,7,8 vs classes 1,4,5,6,9) because the two partitions are more similar to each other. This makes OOD detection more difficult than the 0-4 vs 5-9 split.

### E.7    Vanilla and ERD Ensembles with different architectures

In this section we present OOD detection results for Vanilla and ERD ensembles with different architecture choices, and note that the better performance of our method is maintained across model classes. Moreover, we observe that ERD benefits from employing more complex models, like the WideResNet.

Table 11: Results with three different architectures for Vanilla and ERD ensembles. All ensembles comprise 5 models. For the corruption data sets, we report for each metric the average taken over all corruptions (A), and the value for the worst-case setting (W). The evaluation metrics are computed on the unlabeled set.

| | | VGG16 | | ResNet20 | | WideResNet-28-10 | |
| | | Vanilla Ensembles | ERD | Vanilla Ensembles | ERD | Vanilla Ensembles | ERD |
| ID data | OOD data | | | AUROC ↑ / TNR@95 ↑ | | | |
|---|---|---|---|---|---|---|---|
| SVHN | CIFAR10 | 0.97 / 0.88 | 0.99 / 0.94 | 0.97 / 0.88 | 0.99 / 0.97 | 0.96 / 0.86 | 1.00 / 0.99 |
| CIFAR10 | SVHN | 0.88 / 0.69 | 1.00 / 1.00 | 0.92 / 0.78 | 1.00 / 1.00 | 0.94 / 0.81 | 1.00 / 1.00 |
| SVHN[0:4] | SVHN[5:9] | 0.89 / 0.60 | 0.93 / 0.63 | 0.92 / 0.69 | 0.94 / 0.66 | 0.91 / 0.62 | 0.96 / 0.78 |
| CIFAR10[0:4] | CIFAR10[5:9] | 0.74 / 0.29 | 0.91 / 0.63 | 0.80 / 0.39 | 0.91 / 0.66 | 0.80 / 0.35 | 0.94 / 0.71 |
| CIFAR10 | CIFAR10-C sev 2 (A) | 0.66 / 0.17 | 0.94 / 0.79 | 0.68 / 0.20 | 0.96 / 0.86 | 0.69 / 0.18 | 0.98 / 0.90 |
| CIFAR10 | CIFAR10-C sev 2 (W) | 0.51 / 0.05 | 0.68 / 0.19 | 0.51 / 0.05 | 0.68 / 0.19 | 0.51 / 0.05 | 0.84 / 0.35 |
| CIFAR10 | CIFAR10-C sev 5 (A) | 0.80 / 0.41 | 0.99 / 0.96 | 0.84 / 0.49 | 1.00 / 0.99 | 0.84 / 0.47 | 1.00 / 1.00 |
| CIFAR10 | CIFAR10-C sev 5 (W) | 0.58 / 0.10 | 0.95 / 0.72 | 0.60 / 0.10 | 0.98 / 0.86 | 0.59 / 0.09 | 0.99 / 0.97 |
| | Average | 0.75 / 0.40 | 0.92 / 0.73 | 0.78 / 0.45 | 0.93 / 0.77 | 0.78 / 0.43 | 0.96 / 0.84 |

## E.8 Impact of the ensemble size and of the choice of arbitrary label

In this section we show OOD detection results with our method using a smaller number of models for the ensembles. We notice that the performance is not affected substantially, indicating that the computation cost of our approach could be further reduced by fine-tuning smaller ensembles.

Table 12: Results obtained with smaller ensembles for ERD. All numbers are averages over 3 runs, where we use a different set of arbitrary labels for each run to illustrate our method's stability with respect the choice of labels to be assigned to the unlabeled set. We note that the standard deviations are small ($\sigma \leq 0.01$ for the AUROC values and $\sigma \leq 0.08$ for the TNR@95 values).

| | | K=2 | | K=3 | | K=4 | |
| | | ERD | ERD++ | ERD | ERD++ | ERD | ERD++ |
| ID data | OOD data | | | AUROC ↑ / TNR@95 ↑ | | | |
|---|---|---|---|---|---|---|---|
| SVHN | CIFAR10 | 0.99 / 0.98 | 0.99 / 0.99 | 0.99 / 0.98 | 1.00 / 0.99 | 0.99 / 0.98 | 1.00 / 0.99 |
| CIFAR10 | SVHN | 1.00 / 1.00 | 1.00 / 1.00 | 1.00 / 1.00 | 1.00 / 1.00 | 1.00 / 1.00 | 1.00 / 1.00 |
| CIFAR100 | SVHN | 1.00 / 1.00 | 1.00 / 1.00 | 1.00 / 1.00 | 1.00 / 1.00 | 1.00 / 1.00 | 1.00 / 1.00 |
| SVHN[0:4] | SVHN[5:9] | 0.95 / 0.69 | 0.94 / 0.68 | 0.95 / 0.73 | 0.95 / 0.75 | 0.96 / 0.76 | 0.96 / 0.77 |
| CIFAR10[0:4] | CIFAR10[5:9] | 0.89 / 0.55 | 0.92 / 0.58 | 0.89 / 0.57 | 0.94 / 0.70 | 0.90 / 0.57 | 0.95 / 0.73 |
| CIFAR100[0:49] | CIFAR100[50:99] | 0.81 / 0.40 | 0.82 / 0.43 | 0.81 / 0.41 | 0.84 / 0.44 | 0.81 / 0.41 | 0.84 / 0.44 |
| | Average | 0.94 / 0.77 | 0.95 / 0.78 | 0.94 / 0.78 | 0.95 / 0.81 | 0.94 / 0.79 | 0.96 / 0.82 |

**Impact of the choice of arbitrary labels.** Furthermore, we note that in the table we report averages over 3 runs of our method, where for each run we use a different subset of $\mathcal{Y}$ to assign arbitrary labels to the unlabeled data. We do this in order to assess the stability of ERD ensembles to the choice of the arbitrary labels and notice that the OOD detection performance metrics do not vary significantly. Concretely, the standard deviations are consistently below $0.01$ for all data sets for the AUROC metric, and below $0.07$ for the TNR@95 metric.

## F Medical OOD detection benchmark

The medical OOD detection benchmark is organized as follows. There are four training (ID) data sets, from three different domains: two data sets with chest X-rays, one with fundus imaging and one with histology images. For each ID data set, the authors consider three different OOD scenarios:

1. Use case 1: The OOD data set contains images from a completely different domain, similar to our category of easy OOD detection settings.

2. Use case 2: The OOD data set contains images with various corruptions, similar to the hard covariate shift settings that we consider in Section E.3.

3. Use case 3: The OOD data set contains images that come from novel classes, not seen during training.

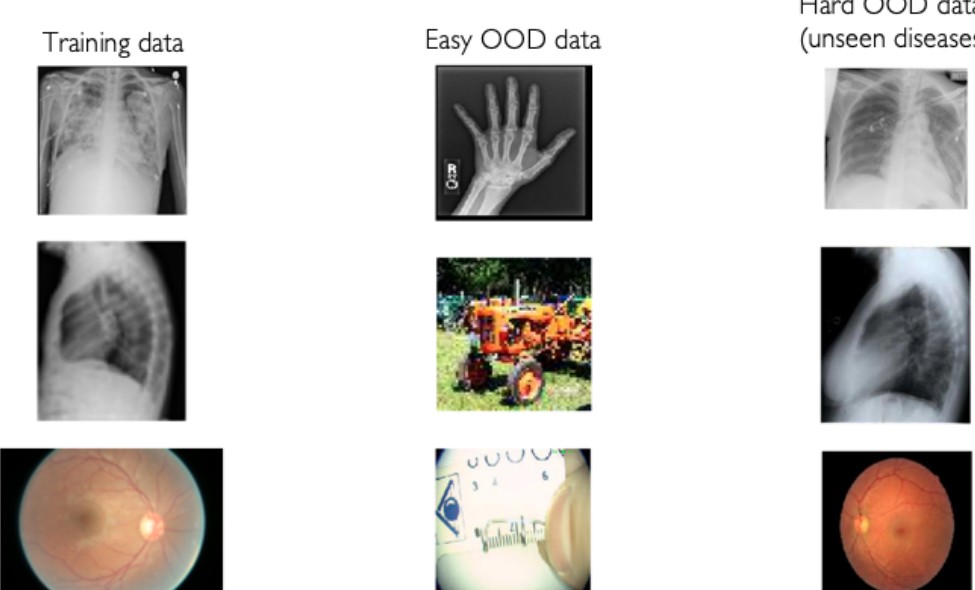

Figure 12: Samples from the medical image benchmark. There are 3 ID data sets containing frontal and lateral chest X-rays and retinal images. Hard OOD samples contain images of diseases that are not present in the training set.

The authors evaluate a number of methods on all these scenarios. The methods can be roughly categorized as follows:

1. Data-only methods: Fully non-parametric approaches like kNN.

2. Classifier-only methods: Methods that use a classifier trained on the training set, e.g. ODIN Liang et al. (2018), Mahalanobis Lee et al. (2018). ERD falls into this category as well.

3. Methods with Auxiliary Models: Methods that use an autoencoder or a generative model, like a Variational Autoencoder or a Generative Adversarial Network. Some of these approaches can be expensive to train and difficult to optimize and tune.

We stress the fact that for most of these methods the authors use (known) OOD data during training. Oftentimes the OOD samples observed during training come from a data set that is very similar to the OOD data used for evaluation. For exact details regarding the data sets and the methods used for the benchmark, we refer the reader to Cao et al. (2020).

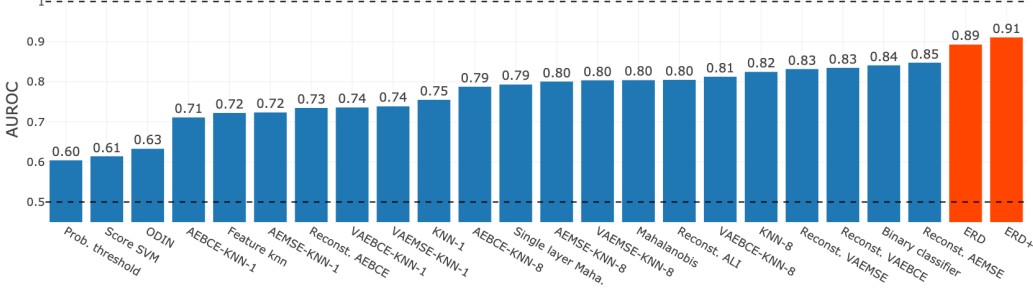

Figure 13: AUROC averaged over all scenarios in the medical OOD detection benchmark Cao et al. (2020). The values for all the baselines are computed using code made available by the authors of Cao et al. (2020). Notably, most of the baselines assume oracle knowledge of OOD data at training time.

In addition, in Figure 14 we present the average taken over only the novel-class settings in the medical benchmark. We observe that the performance of all methods is drastically affected, all of them performing much worse than the average presented in Figure 13. This stark decrease in AUROC and TNR@95 indicates that novelty detection is indeed a challenging task for OOD detection methods even in realistic settings. Nevertheless, our method maintains a better performance than the baselines.

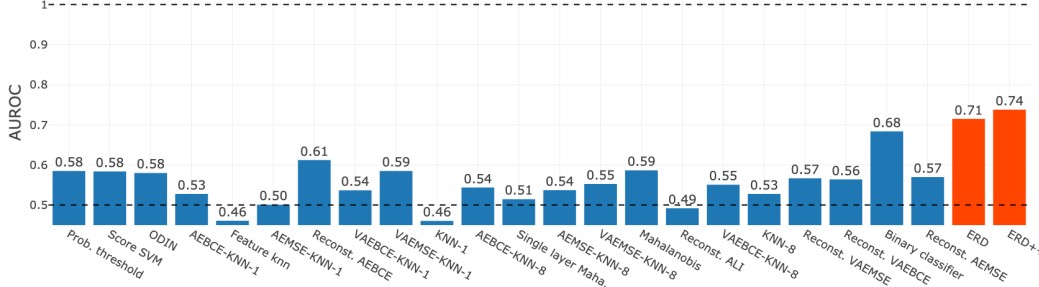

Figure 14: AUROC averaged over the novel-class scenarios in the medical OOD detection benchmark Cao et al. (2020), i.e. only use case 3.

In Figures 15, 16, 17 we present AUROC and AUPR (Area under the Precision Recall curve) for ERD for each of the training data sets, and each of the use cases. Figure 13 presents averages over all settings that we considered, for all the baseline methods in the benchmark. Notably, ERD performs well consistently across data sets. The baselines are ordered by their average performance on all the settings (see Figure 13).

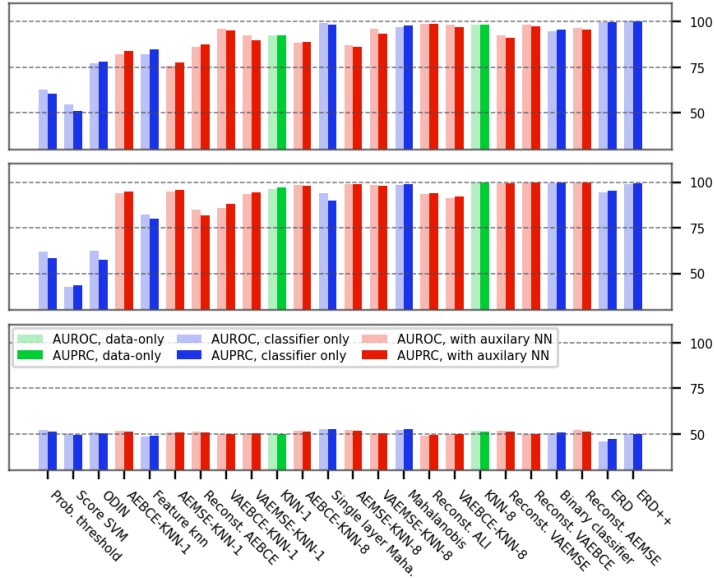

Figure 15: Comparison between ERD and the various baselines on the NIH chest X-ray data set, for use case 1 (top), use case 2 (middle) and use case 3 (bottom). Baselines ordered as in Figure 13.

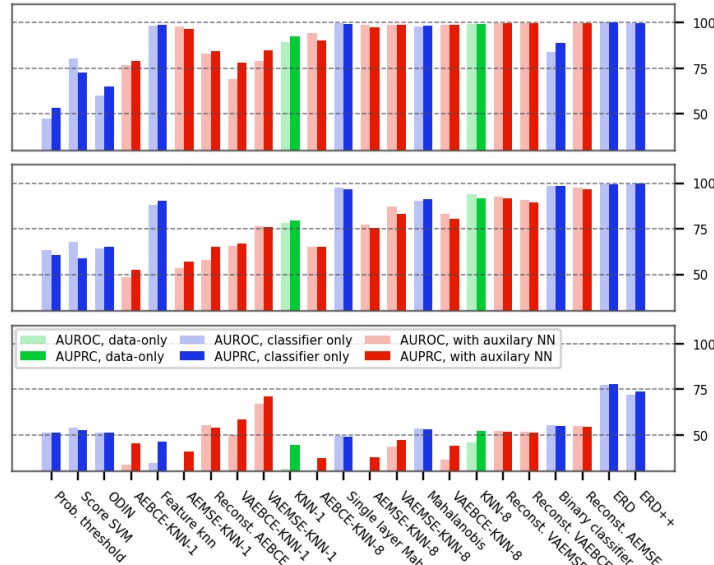

Figure 16: Comparison between ERD and the various baselines on the PC chest X-ray data set, for use case 1 (top), use case 2 (middle) and use case 3 (bottom). Baselines ordered as in Figure 13.

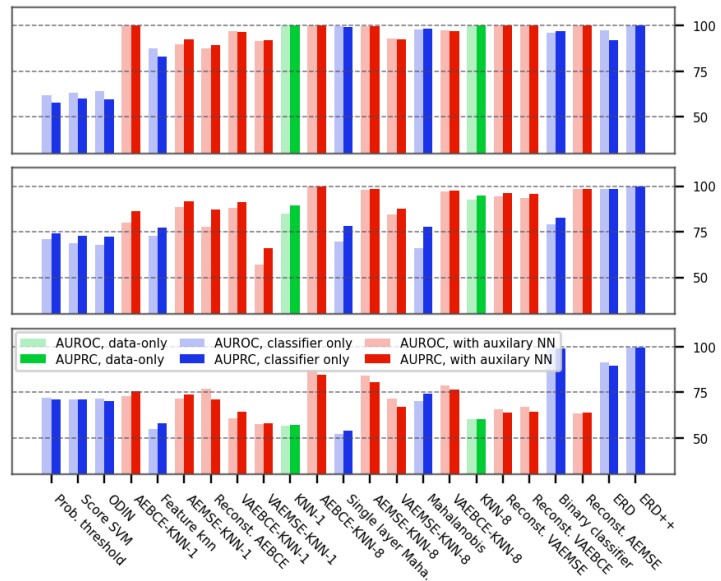

Figure 17: Comparison between ERD and the various baselines on the DRD fundus imaging data set, for use case 1 (top), use case 2 (middle) and use case 3 (bottom). Baselines ordered as in Figure 13.

For all medical benchmarks, the unlabeled set is balanced, with an equal number of ID and OOD samples (subsampling the bigger data set, if necessary). We use the unlabeled set for evaluation.

## G  EFFECT OF LEARNING RATE AND BATCH SIZE

We show now that our method ERD is not too sensitive to the choice of hyperparameters. We illustrate this by varying the learning rate and the batch size, the hyperparameters that we identify as most impactful. As Figure 18 shows, many different configurations lead to similar OOD detection performance.

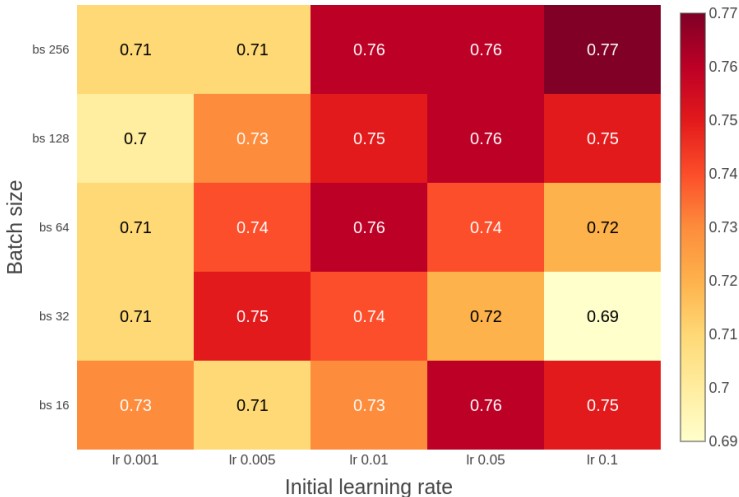

Figure 18: AUROCs obtained with an ensemble of WRN-28-10 models, as the initial learning rate and the batch size are varied. We used the hardest setting, CIFAR100:0-50 as ID, and CIFAR100:50-100 as OOD.

## H    ADDITIONAL FIGURE SHOWING THE DEPENDENCE ON THE UNLABELED SET CONFIGURATION

The configuration of the unlabeled set (i.e. the size of the unlabeled set, the ratio of OOD samples in the unlabeled set) influences the performance of our method, as illustrated in Figure 6b. Below, we show that the same trend persists for different data sets too, e.g. when we consider CIFAR10 as ID data and SVHN as OOD data.

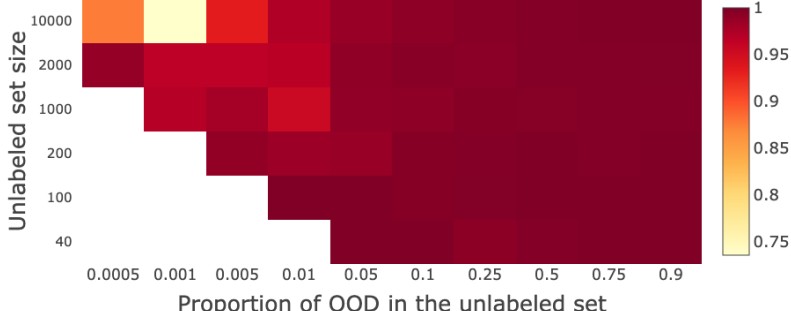

Figure 19: The AUROC of ERD as the number and proportion of ID (CIFAR10) and OOD (SVHN) samples in the unlabeled set are varied.

## I    LEARNING CURVES FOR OTHER DATA SETS

In addition to Figure 5, we present in this section learning curves for other data sets as well. The trend that persists throughout all figures is that the arbitrary label is learned first on the unlabeled OOD data. Choosing a stopping time before the validation accuracy starts to deteriorate prevents the model from fitting the arbitrary label on unlabeled ID data.

**Impact of near OOD data on training ERD ensembles.**    The learning curves illustrated in Figure 20 provide insight into what happens when the OOD data is similar to the ID training samples and the impact that has on training the proposed method. In particular, notice that for CIFAR10[0-4] vs CIFAR10[5-9] in Figure 20d, the models require more training epochs before reaching an accuracy on unlabeled OOD samples of 100%. The learning of the arbitrary label on the OOD samples is delayed by the fact that the ID and OOD data are similar, and hence, the bias of the correctly labeled training set has a strong effect on the predictions of the models on the OOD inputs. Since we

early stop when the validation accuracy starts deteriorating (e.g. at around epoch 8 in Figure 20d), we end up using models that do not interpolate the arbitrary label on the OOD samples. Therefore, the ensemble does not disagree on the entirety of the OOD data in the unlabeled set, which leads to lower OOD detection performance. Importantly, however, our empirical evaluation reveals that the drop in performance for ERD ensembles is substantially smaller than what we observe for other OOD detection methods, even on near OOD data sets.

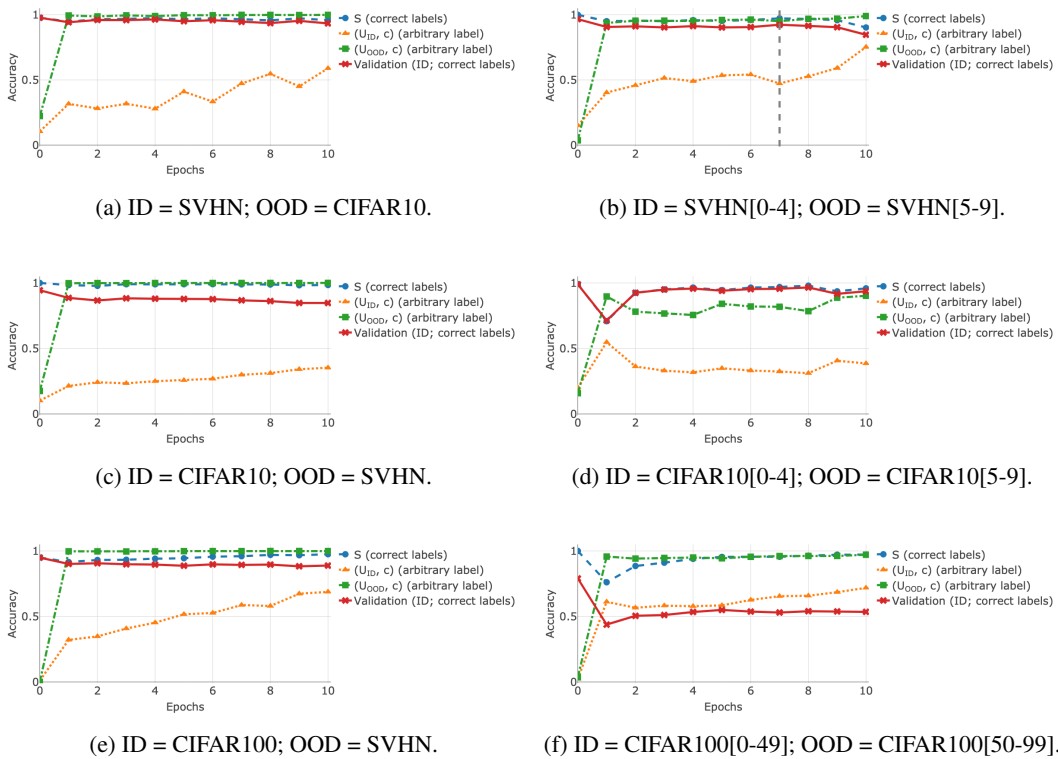

Figure 20: Accuracy measured while fine-tuning a model pretrained on $S$ (epoch 0 indicates values obtained with the initial pretrained weights). The samples in $(U_{\text{OOD}}, c)$ are fit first, while the model reaches high accuracy on $(U_{\text{ID}}, c)$ much later. We fine-tune for at least one epoch and then early stop when the validation accuracy starts decreasing.

## J    EVOLUTION OF DISAGREEMENT SCORE DURING FINE-TUNING

In this section we illustrate how the distribution of the disagreement score changes during fine-tuning for ID and OOD data. Thus, we can further understand why the performance of the ERD ensembles is impacted by near OOD data.

Figure 21 reveals that for far OOD data (the left column) the disagreement scores computed on OOD samples are well separated from the disagreement scores on ID data (note that disagreement on OOD data is so concentrated around the maximum value of 2 that the boxes are essentially reduced to a line segment). On the other hand, for near OOD data (the right column) there is sometimes significant overlap between the disagreement scores on ID and OOD data, which leads to the slightly lower AUROC values that we report in Table 2.

The figures also illustrate how the disagreement on the ID data tends to increase as we fine-tune the ensemble for longer, as a consequence of the models fitting the arbitrary labels on the unlabeled ID samples. Conversely, in most instances one epoch suffices for fitting the arbitrary label on the OOD data.

We need to make one important remark: While in the figure we present disagreement scores for the ensemble obtained after each epoch of fine-tuning, we stress that the final ERD ensemble need not be selected among these. In particular, since each model for ERD is early stopped separately,

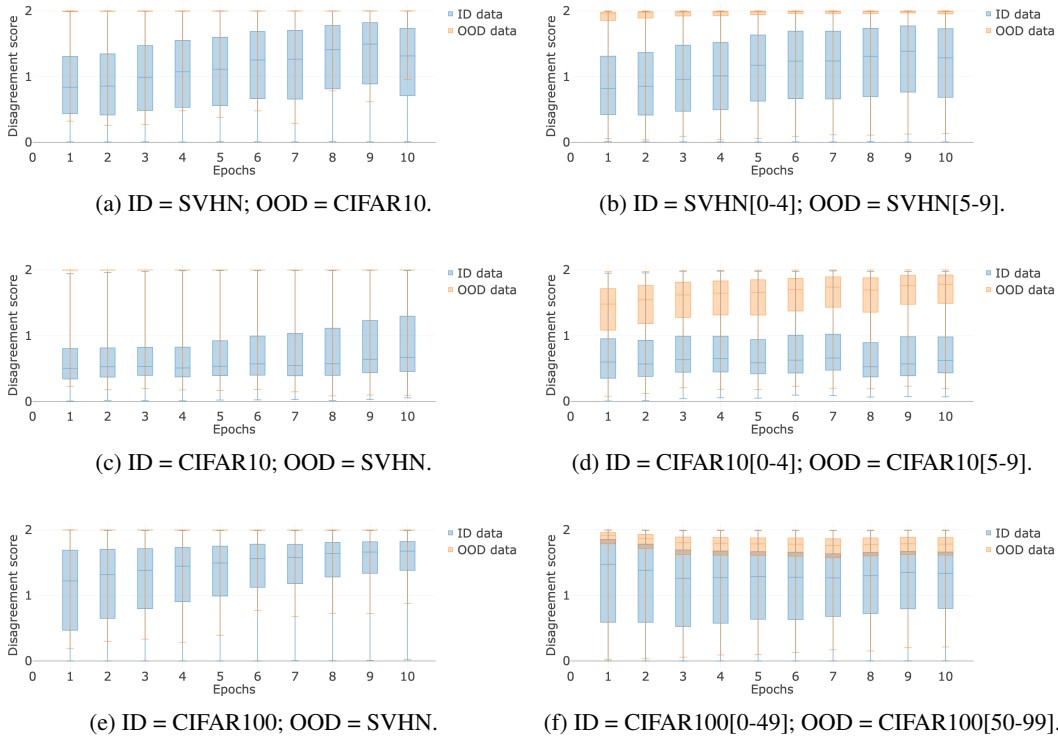

(a) ID = SVHN; OOD = CIFAR10.

(b) ID = SVHN[0-4]; OOD = SVHN[5-9].

(c) ID = CIFAR10; OOD = SVHN.

(d) ID = CIFAR10[0-4]; OOD = CIFAR10[5-9].

(e) ID = CIFAR100; OOD = SVHN.

(f) ID = CIFAR100[0-49]; OOD = CIFAR100[50-99].

Figure 21: The distribution of the disagreement score measured during fine-tuning on ID and OOD data (blue and orange boxes, respectively). The box indicates the lower and upper quartiles of the distribution, while the middle line represents the median and the whiskers show the extreme values. Notice that the distributions of the scores are easier to distinguish for far OOD data (left column), and tend to overlap more for near OOD settings (right column).

potentially at a different iteration, it is likely that the ERD ensemble contains models fine-tuned for a different number of iterations. Since we select the ERD ensembles from a strictly larger set, the final ensemble selected by the our proposed approach will be at least as good at distinguishing ID and OOD data as the best ensemble depicted in Figure 21.

