# OpenReview forum: "Novelty detection using ensembles with regularized disagreement"
_ICLR.cc/2022/Conference — ICLR 2022 Submitted_

### Official Review · Reviewer_VFKa · 2021-10-29

**Correctness:** 3
**Technical Novelty And Significance:** 3
**Empirical Novelty And Significance:** 3
**Recommendation:** 6
**Confidence:** 4

**Main Review:**

1. Problem setting
- In the introduction, the authors mentioned that we have access to the unlabeled test data that includes ID and OOD samples (and using this for training).
- It means that we already have access to the OOD and this may not be applicable to discovering new classes in the inference time that we have never seen before.
- OOD means that the samples from the out of the training set distribution. However, if we use some OOD samples during training, those samples are no longer OOD.
- If we use this for a testing batch, we should retrain the model with a new testing batch. This may be not practical in some online settings.
- It would be good to mention the limitations clearly. Like changing the title as "transductive novelty detection instead of just novelty detection)

2. Number of unlabeled samples
- It seems like the number of unlabeled samples are also important for fine-tuning.
- This is because if we have too small a number of unlabeled samples, it will not provide enough randomness to the ensemble.
- It would be great if we have this additional analysis with different numbers of unlabeled data. (including very small number of unlabeled samples)

3. Number of label categories
- How about if the problem is binary classification?
- In that case, K should be 2. (We can increase K but it does not have an impact).
- In that case, I think the randomness that the ensemble can get is not enough.

4. Validation set
- Based on Section 3, it seems like the validation set is the subset of ID.
- In that case, the best validation performance would be "before" fine-tuning. Because, before fine-tuning, we only use the ID samples to train the model.
- With newly labeled unlabeled data, if the number of labeled categories is large, most of them are incorrectly labeled. Therefore, I think the validation error should be decreased from the beginning.
- The authors said that they wait for 1 epoch. However, it is not sure whether this rule of thumbs can be applicable to the general datasets.

5. Ablation studies
- It would be also good if the methods can work with different K. Especially, if the label categories are small (e.g., binary classification).
- It would be good if the methods can work with small unlabeled sets like 10, 20.
- It would be good if the methods can work with different OOD data ratios in the unlabeled data. (Like 1%, 2% OOD and 99%, 98% ID samples in the test set)

**Summary Of The Paper:**

The authors proposed a novel transductive novelty detection method using the disagreements of the ensemble models.
More specifically, the authors directly utilize the unlabeled test set samples, providing different labels for those unlabeled samples, and training multiple models with fine-tuning. Then, this framework tries to identify the OOD samples using the disagreement of those samples.
The authors provide some promising experimental results to discover the OOD samples in transductive settings.

**Summary Of The Review:**

Strengths
- The proposed method is intuitively making sense.
- The main idea "Introducing the randomness to multiple models using different label assignments to the unlabeled datasets" would be promising to discover the samples which are not presented in training set.
- The experimental results are clear and concrete. The baselines are somewhat complete and the comparisons seem fair.

Weakness
- We need another training procedure for different testing set which is less practical.
- Experimental settings should be more diverse. More ablation studies are needed.
- Robustness of the model training is one of the biggest concerns about this paper. Relying on the validation error for early stopping to avoid overfitting to the ID unlabeled samples seem somewhat heuristic.

--------------------------------------------------

I am appreciate to get the detailed responses from the authors in this rebuttal.
I carefully read all the reviews from other reviewers and authors response.
However, there are still remaining concerns.

1. Practicality of the settings
- Not only me, other reviewers also concerned on the setting of this paper.
- Also, the authors did not answer about re-training problem with new test set. Retraining again and again for new test set would be very painful.

2. Fundamental limitations of the methods
- The robustness of the proposed method is highly related to the diversity of the submodels for the ensemble.
- However, this diversity is significantly limited if the number of unlabeled data is small or the problem is the binary classification.
- Even though the authors provide some interesting results with this setting, the fundamental limitations of the model itself remain concerns.

3. Early stopping
- As I mentioned above, the best validation performance would be the model "before the fine-tuning" because there is no distribution mismatch between training and validation.
- It is unclear how fundamentally address this problem even after reading the rebuttals.

Therefore, even though the authors provide some additional experiments (thank you), I am going to stay on my score (6) and cannot provide more than the score (6).

---

> ### Author Response · Authors · 2021-11-12
> **Response to Reviewer VFKa (1/2)**
>
> We thank Reviewer VFKa for appreciating the novelty of our approach and for providing valuable feedback for improving our manuscript. In what follows, we address the points raised by Reviewer VFKa and kindly encourage them to express any outstanding concerns they may have after having read our response.
>
> >OOD means that the samples from the out of the training set distribution. However, if we use some OOD samples during training, those samples are no longer OOD. If we use this for a testing batch, we should retrain the model with a new testing batch. This may be not practical in some online settings.
>
> **Semi-supervised novelty detection is a valid OOD detection setting.** We define OOD data in contrast to the labeled ID training data which is already known to human experts (e.g. known diseases) and we would like to stress that in the semi-supervised novelty detection (SSND) setting, we do not know which unlabeled samples are OOD or ID. Even when OOD samples are available in the unlabeled set, the problem of precisely detecting them remains challenging, and we show that prior SSND approaches (e.g. nnPU, MCD etc) fail to leverage the unlabeled set effectively in order to achieve good OOD detection performance.
>
> As Reviewer VFKa pointed out, SSND approaches do not lend themselves well to online settings. However, there exist numerous real-world applications where a small delay necessary to collect a batch of test data is a small price to pay in exchange for significantly better OOD detection, as we argue in the Limitations paragraph in Section 5. We note that the other semi-supervised novelty detection works in the literature also provide examples motivating the setting and its practical applicability (see e.g. Blanchard et al, Munoz-Mari et al, Liu et al, Yu et al etc). Finally, we believe that a method that can benefit offline applications by greatly improving upon the state-of-the-art may still be of interest to practitioners in those fields, despite not being fully tailored to other related settings (e.g. real-time OOD detection). We would be curious to know if Reviewer VFKa shares our view on this matter and kindly invite them to express any outstanding concerns they may still have.
>
> >It would be great if we have this additional analysis with different numbers of unlabeled data. (including very small number of unlabeled samples)
>
> **Impact of changing the configuration of the unlabeled set.** Indeed, our approach benefits from a large enough unlabeled set, with a sufficient number of OOD samples. However, in Figure 6b (and also in Appendix E.4 and Appendix H) we show that our method maintains a good OOD detection performance for a wide range of unlabeled set size and a broad spectrum of OOD to ID ratios in the unlabeled set. We would be curious to hear if these ablations address the points raised by Reviewer VFKa.
>
> >How about if the problem is binary classification? [...] In that case, I think the randomness that the ensemble can get is not enough.
>
> **Varying the ensemble size.** We have added in Appendix E.8 a comprehensive analysis of our method’s performance when we vary the ensemble size and note that there is no substantial drop in performance compared to a 5-model ensemble.
>
> Moreover, we would like to point out that the medical data sets on which we evaluate our approach (see Figure 6a and Appendix F) do indeed have binary labels. However, this does not prevent our 2-model ERD ensembles from achieving consistently better OOD performance compared to the baselines.
>
> >The best validation performance would be "before" fine-tuning. Because, before fine-tuning, we only use the ID samples to train the model. [...] I think the validation error should be decreased from the beginning. The authors said that they wait for 1 epoch. However, it is not sure whether this rule of thumbs can be applicable to the general datasets. [...] Relying on the validation error for early stopping to avoid overfitting to the ID unlabeled samples seem somewhat heuristic.
>
> **Theoretical and empirical justification for early stopping.** We would like to stress that using early stopping to prevent fitting the incorrectly labeled ID samples is not simply a heuristic, but rather a choice that stems from rigorous theoretical arguments, as we argue in Section 3.2. In particular, in Appendix A we extend existing results to show that there exists indeed an optimal stopping time when the arbitrary label is only fit on the OOD data, but not on the ID samples in the unlabeled set. We draw on this key insight to design our OOD detection algorithm. The choice to wait for one epoch before early stopping is only necessary when fine-tuning a pretrained model, as astutely pointed out by Reviewer VFKa. Our extensive quantitative experiments as well as the qualitative evidence in Appendix I and J indicate that this procedure is suitable for a wide range of far and near OOD data sets.

---

> > ### Author Response · Authors · 2021-11-12
> > **Response to Reviewer VFKa (2/2)**
> >
> > > Robustness of the model training is one of the biggest concerns about this paper.
> >
> > **Ablation studies to show the stability of the approach.** We would like to stress that we have conducted extensive ablation studies in order to reveal the limitations of our method and to rule out concerns regarding the brittleness of the approach. Concretely, we analyze how the performance of our method varies 1) with the ensemble size (Appendix E.8); 2) with a different assignment of the arbitrary label to the unlabeled set (Appendix E.8); 3) with the model architecture (Appendix E.7); 4) with a different composition of the unlabeled set (Figure 6b and Appendix E.4 and H); 5) with hyperparameters such as learning rate and batch size that are not optimally tuned (Appendix G); and finally 6) with OOD data sets that are increasingly more difficult to detect (Appendix E.3 and F and also the results in Tables 2 and 5). We identify two main limitations of our approach that we discuss openly in the main text. Ensembles trained with early stopping i) tend to disagree less on OOD data when there are few OOD samples in the unlabeled set; and ii) tend to have poorer performance when ID and OOD data are similar (e.g. CIFAR100[0-49] vs CIFAR100[50-99]). We would be happy to hear any suggestions from Reviewer VFKa that would help extend our comprehensive experimental study in order to address the concerns they may still have regarding the robustness of our approach.

---

> > > ### Comment · Reviewer_VFKa · 2021-11-29
> > > **Thank you for the response.**
> > >
> > > Thank you for providing the detailed responses from the authors in this rebuttal. I carefully read all the reviews from other reviewers and authors response. However, there are still remaining concerns.
> > >
> > > 1. Practicality of the settings
> > >
> > > Not only me, other reviewers also concerned on the setting of this paper.
> > > Also, the authors did not answer about re-training problem with new test set. Retraining again and again for new test set would be very painful.
> > >
> > > 2. Fundamental limitations of the methods
> > >
> > > The robustness of the proposed method is highly related to the diversity of the submodels for the ensemble.
> > > However, this diversity is significantly limited if the number of unlabeled data is small or the problem is the binary classification.
> > > Even though the authors provide some interesting results with this setting, the fundamental limitations of the model itself remain concerns.
> > >
> > > 3. Early stopping
> > >
> > > As I mentioned above, the best validation performance would be the model "before the fine-tuning" because there is no distribution mismatch between training and validation.
> > > It is unclear how fundamentally address this problem even after reading the rebuttals.
> > > Therefore, even though the authors provide some additional experiments (thank you), I am going to stay on my score (6) and cannot provide more than the score (6).

---

> > > > ### Author Response · Authors · 2021-11-30
> > > > **Response to Reviewer VFKa**
> > > >
> > > > We thank Reviewer VFKa for responding to our rebuttal.
> > > >
> > > > >  Practicality of the settings
> > > >
> > > > As we pointed out before, we do not introduce this setting. This is a well-established problem, that appears under various different names in the literature: semi-supervised novelty detection (Scott et al, 2008; Blanchard et al 2010, Munoz-Mari et al, 2010), learning with augmented classes (Zhang et al 2020; Da et al 2014 etc), positive-unlabeled learning (Du Plessis et al 2014; Kiryio et al, 2017 etc). All of these works assume the exact same setting as we do, and we would be curious to know why Reviewer VFKa finds this unreasonable.
> > > >
> > > > Indeed adapting at inference-time requires more resources than simply performing inference. However, we show that with our approach, this comes with substantial gains in OOD detection performance, which can be of great importance to many practical applications. Moreover, test-time adaptation is also a well-established paradigm, both in novelty detection (see SSND references) and in other related problems such as domain adaptation.
> > > >
> > > > > Fundamental limitations of the methods
> > > >
> > > > We show in Appendix E.8 that ERD ensembles maintain their good detection performance even when the ensemble size is only 2. Similarly, the improvement observed on the medical data sets comes also with 2-model ensembles, since the ID data has binary labels in those situations. Therefore, these findings stand to contradict the point that the "diversity [of ERD ensembles] is significantly limited if the number of unlabeled data is small or the problem is the binary classification".
> > > >
> > > > > Early stopping
> > > >
> > > > We include a thorough justification for using early-stopping regularization, that includes both theoretical and empirical arguments. Waiting for one epoch of fine-tuning is a heuristic that is only needed for the fine-tuned models. Its role is that it allows for the models to change at all after observing the unlabeled set; otherwise, the unlabeled data would not have any impact on the final ensembles. This heuristic is not necessary when training ensembles from random initialization. Moreover, our theoretical justification of early-stopping does not rely on using this procedure either.

---

### Official Review · Reviewer_QNSM · 2021-11-01

**Correctness:** 3
**Technical Novelty And Significance:** 2
**Empirical Novelty And Significance:** 2
**Recommendation:** 3
**Confidence:** 4

**Main Review:**

Strength:
1. Strong empirical Results on Hard/Near OOD settings.
2.Theoretical insights on the benefits of training with early stopping and the usage of artificial labels
Weakness:
1. The setting of this framework is too restrictive and thus makes the proposed method less practical for real life applications.  For instance, the proposed method requires a carefully selected validation set with ID samples and an unlabeled test set where half of the samples are ID and the other half are OODs. In practice, it is difficult, and somes impossible, to predefine the OOD samples and thus it is not likely to prepare such a high-quality validation set.  Also, it is necessary to investigate how the distributions of the validation set will affect the final performance. Moreover,  as the OOD samples are relatively rare to obtain, the resulting validation set could be highly imbalanced. Hence, using the validation accuracy to guide the model learning is definitely not an optimal solution.
2. The experimental comparisons are not fair. The proposed methods are based on esnemble of 5 networks and this will linearly increase the computational complexity. Thus, it is necessary to have a more fair ``Vanilla Ensembles" by training all the baseline method 5 times with some randmization techniques, such as random sampling on the training set (bagging).
3. The training pipeline of the proposed method is complicated and it is not clear how will each part affect the final performance. For instance, the proposed method finetune K models, where each model will choose one different artificial label on the unlabled set. How will the choice of this artificial label affect the final performance?

**Summary Of The Paper:**

The authors develop an ensemble-based procedure for semi-supervised novelty detection (SSND). It utilizes a mixture of unlabeled ID and OOD samples to perform on near OOD data. A regularization technique is further used  to promote diversity on the OOD data while preserving agreement on ID data.

**Summary Of The Review:**

The proposed method have some empirical results on the Hard/Near OOD. However, this relies heavily on the carefully selected  and high-quality validation sets.  Also, the experimental comparison is not fair.

---

> ### Author Response · Authors · 2021-11-12
> **Response to Reviewer QNSM (1/2)**
>
> We thank Reviewer QNSM for appreciating our work and for the provided feedback. In what follows, we address the comments made by Reviewer QNSM and invite them to express any further concerns they may have after having read our response.
>
> > The setting of this framework is too restrictive and thus makes the proposed method less practical for real life applications. For instance, the proposed method requires a carefully selected validation set with ID samples and an unlabeled test set where half of the samples are ID and the other half are OODs. In practice, it is difficult, and somes impossible, to predefine the OOD samples and thus it is not likely to prepare such a high-quality validation set.
>
> **Clarification regarding the semi-supervised novelty detection setting.** We believe that an important appeal of the semi-supervised novelty detection (SSND) setting is precisely that it relieves the burden of having to collect a sensible set of OOD data for training. We would like to clarify that SSND approaches like ours propose a way to **adapt** to OOD data that emerges at inference time. Therefore, the unlabeled set is not a static object only prepared once, but instead it is collected continuously from inference-time data. In practice, one can even use a batch of test data as the unlabeled set, in what resembles a transductive learning setting. Thus, we eliminate the necessity of curating a set with known OOD samples that have to be representative of any OOD data that may occur at inference time.
>
> In the limitations paragraph of Section 5 we present in detail how we envision that our method could be applied in a real-world setting. Finally, we refer to other semi-supervised novelty detection works in the literature for further discussions motivating the setting and its practical applicability (see e.g. Blanchard et al, Munoz-Mari et al, Liu et al, Yu et al etc). We would be curious to know if our clarification addresses the concerns of Reviewer QNSM and invite them to express any remaining doubts that they may have regarding the setting.
>
> >Also, it is necessary to investigate how the distributions of the validation set will affect the final performance. Moreover, as the OOD samples are relatively rare to obtain, the resulting validation set could be highly imbalanced.
>
> **Impact of the configuration of the unlabeled set.** Here we assume that Reviewer QNSM meant “unlabeled set” instead of “validation set” and kindly ask them to correct us if our assumption is misguided. Indeed, the performance of our approach depends to some extent on the composition of the unlabeled set, and in particular on the amount of OOD data that is present in it. We discuss this dependence in the limitations paragraph in Section 5 and show how the AUROC of our method changes with different unlabeled set configurations in Figure 6b and in Appendix H. Moreover, in Appendix E.4 we show how our method performs on all data sets when a significantly smaller unlabeled set is used for fine-tuning.
>
> >Hence, using the validation accuracy to guide the model learning is definitely not an optimal solution.
>
> **Motivation for early stopping based on the ID validation accuracy.** When training the ensembles, we want that the models do not learn the wrong label on the ID data. If a model did learn the incorrect labels on the ID data, then the generalization accuracy would deteriorate and we would notice a drop in its finite-sample surrogate, the ID validation accuracy. This makes the accuracy on the ID validation set an ideal early stopping criterion. A more formal justification for using the accuracy on the validation set to decide when to early stop is laid out in detail in Section 3.3 and in Appendix A. Once again, to avoid confusion, we would like to clarify that the validation set contains solely ID samples. We would like to ask Reviewer QNSM to confirm whether our clarification helped address their concerns regarding the use of the ID validation set for an early stopping criterion.

---

> > ### Author Response · Authors · 2021-11-12
> > **Response to Reviewer QNSM (2/2)**
> >
> > >The experimental comparisons are not fair. The proposed methods are based on esnemble of 5 networks and this will linearly increase the computational complexity. Thus, it is necessary to have a more fair ``Vanilla Ensembles" by training all the baseline method 5 times with some randmization techniques, such as random sampling on the training set (bagging).
> >
> > **Comparison with related works.** We are not entirely sure that we follow Reviewer QNSM’s suggestion. We note that we are following the same evaluation protocol that is standard in the OOD detection literature (see e.g. Lakshminarayanan et al, Yu et al, Sastry et al, Lee et al etc), and that is also used by other works that introduce ensemble-based approaches for OOD detection. To the best of our knowledge, we are not aware of any such attempts, in other OOD detection works, to ensemble baselines that are not intended to be used as part of an ensemble. We would like to kindly ask Reviewer QNSM to point us to related works that do indeed perform this kind of experimental analysis.
> >
> > Having said that, we would like to point out that our method maintains its good performance even with ensembles of 2 models, as illustrated in Appendix E.8 of the revised manuscript (former Appendix G.1 in the old version). We also highlight that some of the baselines also employ ensembles (e.g. vanilla ensembles, MCD) while others, such as DPN, OE or Mahalanobis, train a WideResNet-28-10 which contains roughly 3 times more trainable parameters compared to the ResNet20 that we use for our method.
> >
> > >The training pipeline of the proposed method is complicated and it is not clear how will each part affect the final performance. For instance, the proposed method finetune K models, where each model will choose one different artificial label on the unlabled set. How will the choice of this artificial label affect the final performance?
> >
> > **Ablation studies to show the stability of the approach.** We believe that one of the main strengths of our approach lies precisely in its simplicity and stability with respect to hyperparameter choices. We would like to stress that we have conducted extensive ablation studies in order to reveal the limitations of our method and to rule out concerns regarding the brittleness of the approach. Concretely, we analyze how the performance of our method varies 1) with the ensemble size (Appendix E.8); 2) with a different assignment of the arbitrary label to the unlabeled set (Appendix E.8); 3) with the model architecture (Appendix E.7); 4) with a different composition of the unlabeled set (Figure 6b and Appendix E.4 and H); 5) with hyperparameters such as learning rate and batch size that are not optimally tuned (Appendix G); and finally 6) with OOD data sets that are increasingly more difficult to detect (Appendix E.3 and F and also the results in Tables 2 and 5).
> >
> > In particular, we include in the revised manuscript a thorough analysis of the impact of the choice of the arbitrary labels that are assigned to the unlabeled set in Appendix E.8. For each data set we train 3 separate ensembles, each with a different set of arbitrary labels, and notice that the choice of the arbitrary labels has very little to no effect on our method: the standard deviation over the 3 runs is consistently lower than 0.01 for the AUROC metric and lower than 0.07 for the TNR@95 metric, for all the data sets and the ensemble sizes that we have considered. We would be happy to hear any suggestions from Reviewer QNSM that would help extend our comprehensive experimental study in order to address the concerns they may still have regarding our approach.

---

> > > ### Comment · Reviewer_QNSM · 2021-11-27
> > > **Thanks for the response**
> > >
> > > Thanks for the response.  I still do not accept the statement of  ``use a batch of test data as the unlabeled set" .  As for the experimental evaluation, the proposed method will linearly increase the computational complexity. Hence, my suggestion is, you can try to linearly increase the computational complexity of the baseline methods to generate ensembles. A very straightforward way is to use bagging. This will help us to further understand that whether the proposed method will have any advantages.
> > >
> > > Finally, I will keep my original score.

---

> > > > ### Author Response · Authors · 2021-11-27
> > > > **Response to Reviewer QNSM**
> > > >
> > > > Firstly, we would like to thank Reviewer QNSM for taking the time to respond to our rebuttal.
> > > >
> > > > > I still do not accept the statement of ``use a batch of test data as the unlabeled set"
> > > >
> > > > We would like to kindly ask Reviewer QNSM to elaborate on their statement. It is not clear to us why they choose to not accept what is a well-established problem (albeit a lesser-known one), namely transductive/semi-supervised novelty detection. As we stress in the manuscript, this problem goes back at least to 2008 and the seminal work of Blanchard and Scott, and has been tackled several times in the recent OOD detection literature (see e.g. Munoz-Mari et al, 2010; Liu et al, 2018; and Yu et al, 2019 references in our paper). **All** of these works assume the exact same setting as we do, and we would be curious to know why Reviewer QNSM finds this unreasonable.
> > > >
> > > > > Hence, my suggestion is, you can try to linearly increase the computational complexity of the baseline methods to generate ensembles.
> > > >
> > > > As we stress in the previous response, we are not aware of any prior work that ensembles OOD detection methods that are not originally meant to be used in an ensemble. Such an endeavor would pose challenges of its own (e.g. how to OOD score for the ensemble) which can affect the outcome of the experimental comparison profusely. Having said that, a parameter that we can tune in order to address concerns about the computational complexity of our approach is the size of the ERD ensemble, as well as the model architecture used for our ensembles. While tuning both (see Appendix E.7 and E.8) we observe that even with a 2-model ensemble, and even when using a much smaller VGG16 architecture, our method outperforms the baselines (some of which use substantially larger model architectures like WideResNet).

---

### Official Review · Reviewer_iZuG · 2021-11-01

**Correctness:** 3
**Technical Novelty And Significance:** 2
**Empirical Novelty And Significance:** 2
**Recommendation:** 5
**Confidence:** 4

**Main Review:**

Strengths:

- simple idea; easy to understand
- a theoretical justification of the approach is provided (in the appendix)
- good results

Weaknesses:

- The semi-supervised requirement of the approach limits its applicability. The availability of clean labeled data is rare in practical scenarios.

- The main set of experiments assume that the unlabeled data used during training follows the same distribution of the data used for testing.

- The paper does not clearly identify the limitations of the proposed approach and scenarios in which the approach would not work.

The scenarios depicted in Figures 2 and 3, while useful to convey the overall idea behind the methodology, show an ideal case only. What happens if the ID classes present a non negligible overlap, and this overlap is also reflected in the unlabeled ID data? Would the early stopping criterion still be effective to achieve the desired disagreement in the ensemble? When does the proposed technique break? Under which conditions? What happens when the OOD do not belong to specific classes with a well-defined support? What happens when there are fewer OOD data. This latter point is somewhat discussed in the appendix, but it should be covered in the main paper as well.

The data used for Figure 1 should be specified.

The authors need to make it clear in Algorithm 1 that the validation set V is made up of in-distribution data.

How was the ensemble size K=5 chosen?

It's unclear whether the values in Table 2 are averages of multiple runs, as they should be. Statistical significance should also be reported.

In Table 2, for each dataset half of the classes were used to construct the ID data, and half for the OOD data. How was the desired ratio between ID and OOD achieved? Were the classes for OOD data sub-sampled? This is not discussed in the paper.



**Summary Of The Paper:**

The authors introduce a semi-supervised ensemble approach to novelty detection. The main idea consists in generating base classifiers that disagree on the out-of-distribution data (ODD). An early-stop criterion is used to achieve the wanted level of disagreement among the component of the ensemble.

The main assumption is the availability, for training, of clean labeled in-distribution (ID) data, and of unlabeled data which include both ID data and out-of-distribution (OOD) data. The unlabeled data follows the same distribution of the data the ensemble is tested on.

Each base classifier is first trained on the labeled in-distribution (ID) data. For a given base classifier, the unlabeled data is randomly assigned to one of the labels (a different label is chosen for each base classifier), and the classifier is tuned on the resulting labeled data. The tuning is stopped when the optimal tradeoff between high validation accuracy (on inliers) and low training error are achieved. This regularization process is aimed at avoiding fitting the incorrect assigned labels in the unsupervised portion of the training data and to achieve disagreement among the base classifiers on the OOD data.

**Summary Of The Review:**

A reasonable, simple, and interesting approach for anomaly detection with ensembles. The main weaknesses are the reliability on clean ID data and the lack of a discussion and analysis of the approach limitations.

---

> ### Author Response · Authors · 2021-11-12
> **Response to Reviewer iZuG (1/2)**
>
> We thank Reviewer iZuG for finding our approach interesting and for the feedback that they have provided. We incorporate their minor suggestions in the revised version of our manuscript and address their other comments in detail in what follows. We encourage Reviewer iZuG to express any concerns that they may still have after having read our response.
>
> >The semi-supervised requirement of the approach limits its applicability. The availability of clean labeled data is rare in practical scenarios.
>
> **Labeled data as a potential limitation of SSND.** In the semi-supervised novelty detection (SSND) setting, the only labeled data that we assume access to is an ID training set, similar to what is required for supervised learning. To avoid any misunderstanding, we would like to kindly ask Reviewer iZuG to clarify what they meant by the rare availability of clean labeled data being a limitation of the semi-supervised novelty detection setting (this is as much a limitation of any supervised learning setting as well). As we elaborate in section 2.1, the SSND setting has been adopted several times in the past in the context of novelty detection (see e.g. Blanchard et al, Munoz-Mari et al, Liu et al, Yu et al etc). In this work, we simply adopt the same scenario and propose a competitive, state-of-the-art OOD detection method.
>
> >The main set of experiments assume that the unlabeled data used during training follows the same distribution of the data used for testing.
>
> **Clarification regarding evaluating with a different OOD distribution.** We would like to stress that we explicitly avoid fine-tuning the models with OOD data that is different from the test-time OOD data, like in Outlier Exposure, for instance. The reason is simple: we argue that using a curated set of OOD samples for training may give a false sense of security against novel OOD data, which is unjustified since there may always exist unseen classes that remain undetected simply because they are not similar enough to the OOD data that was used to train the models.
>
> Instead, we propose to adapt to inference-time OOD data as it emerges in the semi-supervised novelty detection setting that we adopt in our work. In practice, this can be done by using a test batch as the unlabeled set, in what resembles transductive learning, as we describe in the real-world example in Section 5 (in Appendix E.1 we confirm that evaluating on the unlabeled set instead of a holdout set does indeed have a negligible impact the OOD detection metrics). Moreover, our framework can be applied in domains where relevant unlabeled OOD data is not abundant (e.g. medical applications, agriculture, scientific applications such as biology, archeology etc) and even for problems that go beyond image data. We believe that an important appeal of the SSND setting is precisely that it relieves the burden of having to collect a sensible set of OOD data for training, and instead provides a way to adapt to the exact OOD distribution that occurs at inference time.
>
> Finally, we refer to the other semi-supervised novelty detection works in the literature for further discussions motivating the setting and its practical applicability (see e.g. Blanchard et al, Munoz-Mari et al, Liu et al, Yu et al etc).
>
> >What happens if the ID classes present a non negligible overlap, and this overlap is also reflected in the unlabeled ID data? Would the early stopping criterion still be effective to achieve the desired disagreement in the ensemble?
>
> **The impact of overlapping ID classes.** If the ID data contains overlapping classes, that does not affect the OOD detection performance significantly, as exemplified by the CIFAR100 vs SVHN or CIFAR100 vs CIFAR10 settings, where the ID CIFAR100 data set contains numerous similar classes such as hamster / mouse / squirrel or forest / mountain. Moreover, we note that as long as the models in the ensemble agree to predict the same outcome on an ID sample, that sample will be correctly identified as ID, even if the predicted label is wrong (e.g. due to biases in the training procedure).
>
> >When does the proposed technique break? Under which conditions?
>
> **Extending the discussion on limitations.** We would like to stress that we have conducted extensive ablation studies in order to reveal the limitations of our method and to rule out concerns regarding the brittleness of the approach. Concretely, we analyze how the performance of our method varies 1) with the ensemble size (Appendix E.8); 2) with a different assignment of the arbitrary label to the unlabeled set (Appendix E.8); 3) with the model architecture (Appendix E.7); 4) with a different composition of the unlabeled set (Figure 6b and Appendix E.4 and H); 5) with hyperparameters such as learning rate and batch size that are not optimally tuned (Appendix G); and finally 6) with OOD data sets that are increasingly more difficult to detect (Appendix E.3 and F and also the results in Tables 2 and 5). (continues)

---

> > ### Author Response · Authors · 2021-11-12
> > **Response to Reviewer iZuG (2/2)**
> >
> > (continuation) We identify two main limitations of our approach, which we discuss openly in the main text in Section 5. Simply put, our approach breaks when the models do not fit the arbitrary label on the unlabeled OOD data before the early stopping time (i.e. before learning the wrong label on ID data). This outcome can occur, for instance, if there is a stark imbalance between the ID and the OOD samples in the unlabeled set or if the ID and OOD data are so similar that fitting an arbitrary label on OOD samples requires a complex predictor (i.e. a neural network trained for many iterations). We discuss the former case in detail in the limitations paragraph in Section 5 (see Figure 6b) as well as in Appendix H. A thorough experimental analysis of the latter is more difficult since it would require a reasonable metric to quantify the distance between two image data sets, but we can draw some conclusions by looking at the learning curves of the models in the ensemble (see Appendix I). For instance, for CIFAR10[0-4] vs CIFAR10[5-9] where the ID and OOD classes are similar (e.g. cat and deer are among ID and dog and horse are among OOD) we notice that the arbitrary label is not fit on the entire OOD set at the early stopping time at epoch 8, which explains the slight drop in OOD detection performance. However, our experimental study reveals that ERD ensembles continue to produce the desired disagreement in practice, even on particularly hard near OOD data sets e.g. CIFAR10v2 or CIFAR10/CIFAR100 images with low severity corruptions (see Appendix E.3). Finally, in Appendix J we provide an analysis that illustrates how the disagreement scores evolve during training and how their distribution changes between far and near OOD data.
> >
> > >What happens when the OOD do not belong to specific classes with a well-defined support?
> >
> > **Going beyond novelty detection.** Indeed, the problem of robustness to non-iid test data is remarkably broad and, apart from novelty detection, covers topics such as detecting anomalies/outliers that only occur 1-2 times, robustness to adversarial examples, or dealing with data with covariate shift. Ideally, the ultimate goal of the machine learning community is to have a single algorithm that solves all of these problems. Nevertheless, even using specialized methods to tackle each of the tasks separately turns out to be a challenging endeavor that still leaves a lot of room for improvement. For instance, our experiments show that novelty detection is far from being solved by existing methods, which perform poorly especially on more difficult instances of the problem. Furthermore, anomaly detection has its own challenges and Le Lan et al [1] have shown recently that it cannot be solved even with a perfect density model. This finding suggests that anomaly detection algorithms need to operate under strong assumptions that may be too constraining and unjustified for novelty detection.
> >
> > **Justification for the ensemble size.** We pick the ensemble size to be 5 in order to mirror the methodology introduced in Lakshminarayanan et al for Vanilla Ensembles. While throughout the paper we report results for a fixed ensemble size of 5, we note that our method is not particularly sensitive to this hyperparameter. To make this point clear we add in Appendix E.8 a table with results obtained with our method when using smaller ensembles and note that there is no substantial drop in performance.
> >
> > >In Table 2, for each dataset half of the classes were used to construct the ID data, and half for the OOD data. How was the desired ratio between ID and OOD achieved? Were the classes for OOD data sub-sampled?
> >
> > **Selecting ID and OOD classes for near OOD data set.** For the results in the main paper we fix a single partitioning into ID and OOD classes for each data set and indicate it explicitly in Table 2. Note that we sometimes deliberately choose this partition so as to make the OOD detection problem more difficult (e.g. for FashionMNIST). In Appendix E.2 we also present results obtained with our method when the ID classes are sampled at random (the numbers in Table 5 represent averages over 5 draws of the ID and OOD classes). The OOD detection performance of our approach stays largely unchanged for different partitions of the data (the standard deviation of the AUROC over the 5 draws is consistently smaller than 0.01).
> >
> > [1] - Le Lan et al, Perfect density models cannot guarantee anomaly detection, 2021.

---

### Official Review · Reviewer_Etvi · 2021-11-01

**Correctness:** 3
**Technical Novelty And Significance:** 3
**Empirical Novelty And Significance:** 2
**Recommendation:** 6
**Confidence:** 3

**Main Review:**

The paper is generally well-written, however, there is some confusion listed as follows:

1.	How is the ensemble number selected? What is the validation dataset for choosing it?
2.	Some claims are vague in the submission, for example, in section 3.3, the authors claimed that the Maximum discrepancy method (MCD) tends to result in ensembles that do not disagree enough on OOD data and thus subpar performance. Is there any proof or empirical evidence behind such claims?
3.	In figure 5, is it possible that the decrease of validation accuracy is due to the fitting on the labels of the OOD data rather than the noisy in-distribution labels? One quick experiment to verify this is using clean labels for ID data and then fitting on a new OOD dataset with artificial labels to see the accuracy change on the validation dataset.
4.	In figure 5, how is the early stopping tolerance scalar 7 selected? Is there any sensitivity analysis on it?
5.	It is improper to claim that “However, running a grid search to select the right hyperparameters can be more computationally expensive than simply using one run of the training process to select the optimal stopping time”. Since early stopping requires you to do inference per training epoch and it may be more expensive than tuning the hyperparameters using grid search.
6.	How is the proposed method compared to self-supervised-based approaches, such as CSI[1] and SSD[2]? Since these methods do not require ID labels at all, it has fewer requirements than the proposed approach. How is the method compared to different uncertainty scores, such as energy scores [3]?


[1]Jihoon Tack, Sangwoo Mo, Jongheon Jeong, and Jinwoo Shin. Csi: Novelty detection via contrastive learning on distributionally shifted instances. In Advances in Neural Information Processing Systems, pages 11839–11852, 2020

[2]Vikash Sehwag, Mung Chiang, and Prateek Mittal. Ssd: A unified framework for self-supervised outlier detection. In Proceedings of the 9th International Conference on Learning Representations, pages 1–17,2021

[3] Weitang Liu, Xiaoyun Wang, John Owens and Yixuan Li, Energy-based Out-of-distribution Detection, in NeurIPS 2020.



**Summary Of The Paper:**

Even though conventional OOD detection algorithms can distinguish far OOD samples, current methods that can identify near OOD samples require training with labeled data that is very similar to these near OOD samples. In turn, the authors develop a new ensemble-based procedure for semi-supervised novelty detection (SSND) that only utilizes a mixture of unlabeled ID and OOD samples to achieve good detection performance on near OOD data.

**Summary Of The Review:**

Some confusion exists both empirically and theoretically. I recommend weak-reject at this stage.

---

> ### Author Response · Authors · 2021-11-12
> **Response to Reviewer Etvi (1/1)**
>
> We thank Reviewer Etvi for appreciating our work and for the feedback that they have provided. We hope that our responses address the concerns raised by Reviewer Etvi and warmly encourage them to point out anything else they believe is still unreasonable in our manuscript.
>
> **Justification for the ensemble size.** We pick the ensemble size to be 5 in order to mirror the methodology introduced in Lakshminarayanan et al for Vanilla Ensembles. While throughout the paper we report results for a fixed ensemble size of 5, we note that our method is not particularly sensitive to this hyperparameter. To make this point clear we add in Appendix E.8 a table with results obtained with our method when using smaller ensembles and note that there is no substantial drop in performance.
>
> >Some claims are vague [...] the authors claimed that the Maximum discrepancy method (MCD) tends to result in ensembles that do not disagree enough on OOD data and thus subpar performance. Is there any proof or empirical evidence behind such claims?
>
> **Clarifying our claim regarding MCD ensembles.** Indeed, we observe empirically that ensembles trained with MCD are significantly less diverse than our method and provide evidence for this claim in Appendix B (in particular, see Figure 9). We have now added a clarifying remark pointing to the Appendix with the evidence for our claim. We would be happy to further clarify any claims that are ambiguous in our manuscript and kindly invite Reviewer Etvi to point out other examples of vague statements.
>
> > In figure 5, is it possible that the decrease of validation accuracy is due to the fitting on the labels of the OOD data rather than the noisy in-distribution labels?
>
> **Causes of the decrease in validation accuracy.** Provided that the ID and OOD distributions have disjoint support, it is possible to correctly fit the arbitrary label on the OOD data without paying a price in ID generalization performance, if our predictor is sufficiently complex (as is the case for neural networks). Figure 20 in Appendix I shows that the validation accuracy and the accuracy on the ID data in the unlabeled set are indeed correlated across a wide variety of near and far OOD data sets.
>
> >In figure 5, how is the early stopping tolerance scalar 7 selected? Is there any sensitivity analysis on it?
>
> **Early stopping criterion.** As we describe at the end of section 3.3, we select the early stopping time as the epoch with the highest validation accuracy, after having fine-tuned the models for 10 epochs. We also point to Appendix I for more figures that show the sensitivity of the early stopping time varies to changes of the OOD data set.
>
> > It is improper to claim that “However, running a grid search to select the right hyperparameters can be more computationally expensive than simply using one run of the training process to select the optimal stopping time”. Since early stopping requires you to do inference per training epoch and it may be more expensive than tuning the hyperparameters using grid search.
>
> **Cost of tuning early-stopping regularization.** We point out that for all the data sets that we consider, fine-tuning our method for 10 epochs is enough to find a good stopping time. Therefore, we only perform inference 10 times for each model, on a relatively small validation set (a quarter of the size of the training set). This procedure is less costly than training a model AND performing inference for each configuration in the grid search, which scales exponentially with the number of hyperparameters that need to be tuned. We hope that our answer addresses Reviewer Etvi’s question and, if that is not the case, we would like to ask them to clarify what precisely they meant.

---

> > ### Author Response · Authors · 2021-11-12
> > **Response to Reviewer Etvi (1/2)**
> >
> > > How is the proposed method compared to self-supervised-based approaches, such as CSI[1] and SSD[2]? Since these methods do not require ID labels at all, it has fewer requirements than the proposed approach. How is the method compared to different uncertainty scores, such as energy scores [3]?
> >
> > **More related work.** We thank Reviewer Etvi for pointing out these related works. We note that we already include Tack et al in our taxonomy (see the middle cell in the Synthetic OOD A-UND column in Table 1) and show in Table 5 (Appendix E.2) that our method performs better than CSI on near OOD detection, even when CSI uses a labeled training set. We emphasize that the performance of both CSI and SSD decreases substantially when they use an unlabeled ID training set instead. In the revised manuscript, we also add a comparison with the labeled version of SSD in Table 5 in Appendix E.2, which still performs worse than our proposed approach.
> >
> > In Table 2 in the Experiments section, we compare our method against uncertainty-based approaches (e.g. DPN, Vanilla Ensembles) and show that they struggle to correctly identify both near and far OOD samples. Similarly, the method proposed in [3] has a relatively low AUROC on far OOD settings like CIFAR10/CIFAR100 vs SVHN compared to our method (see Tables 4 and 5 in [3]) and we expect that it also underperforms on near OOD data.

---

> > > ### Comment · Reviewer_Etvi · 2021-11-26
> > > **Thanks for the response**
> > >
> > > Thanks for the author's response! It indeed solves some of my confusion. However, considering the limitations proposed by the other reviewers, I will increase the score to 6.
> > >
> > > Thanks,
> > >
> > > Reviewer Etvi

---

### Official Review · Reviewer_7CPC · 2021-11-02

**Correctness:** 4
**Technical Novelty And Significance:** 2
**Empirical Novelty And Significance:** 3
**Recommendation:** 5
**Confidence:** 4

**Main Review:**

-The setting of the paper assumes that a mixture of in- and out-distribution samples is available during training and the authors argue for the validity of this assumption in the conclusion section of the paper. While I agree that this specific setting can come up in real-world applications, I think it is still important to evaluate unseen out-distributions. For example, if I train a model on medical images using their procedure, can the model detect "garbage samples", for example uniform noise, that might result from a technical failure in the image capturing pipeline. Another interesting setting might be to only train a CIFAR10 model on the first 50 CIFAR100 classes in the unlabeled pool available during training and evaluate OOD-detection performance on the last 50 classes from CIFAR100 and other standard benchmarks such as AUROC against SVHN or Uniform and Gaussian Noise. The paper clearly demonstrates that the model is able to beat baseline methods such as nnPU and MCD if the distribution of out-of-distribution samples stays the same between train and test time, however, I think it important to also evaluate robustness with respect to unseen out-of-distributions, especially as methods such as Outlier Exposure have shown that they can generalize to various unseen out-distributions.
It might also be interesting to replace the unlabeled set U with a general image dataset with enough variance, such as a subset of 80 million tiny images, OpenImages or YFCC-100M which all contain some samples of the standard CIFAR10/100 classes but also a vast number of unseen classes. Using such a general dataset might also lead to a better generalization to unseen out-distributions, similar to Outlier Exposure and it would allow to computer AUROC of a CIFAR10 model against all CIFAR100 classes, which is a challenging benchmark that is often looked at in related papers.

-The paper shows strong improvements over a wide variety of baseline methods in terms of AUROC and TNR for several tasks based on SVHN, FMNIST, CIFAR10, CIFAR100 and a medical benchmark.

-The success of the method largely relies on the models in the ensemble having sufficiently different outputs on OOD samples. The authors argue that during training, the model will first first the correctly labeled samples in U_ID before fitting the wrongly labeled ones.
In the appendix, the authors give further theoretical insight and state that under some assumptions, there exists a checkpoint after T iterations where the model fits the correctly labeled samples in U_ID but not the incorrectly labeled ones. In practice, this T value can not be calculated so the final checkpoint is selected based on accuracy on a validation set that only contains in-distribution samples. However, I am not sure how this relates to disagreement on actual out-of-distribution samples, which is arguably the goal of the training scheme to achieve best OOD-performance. In detail, what happens if all models in the ensemble predict the same label c on an out-distribution sample, wouldn't all models still learn this output before we end training? This systematic bias might be a problem if the OOD classes are similar to the in-distribution classes, for example train or busses might be similar to trucks in CIFAR10 and as the method does not seem to contain an explicit way to generate diverse initialization for their fine-tuning scheme.
Figure 9 in the appendix gives hope that one has sufficient disagreement,, however, I am not totally convinced that the first 5 CIFAR10 classes are sufficiently close to the last 5 classes to completely eliminate this concern.

-I liked that the authors show that the method works with varying proportions of OOD in the unlabeled dataset unless U consists of 95% in-distribution samples.

-The authors also propose to use a different disagreement score and from Table 3 it seems like at least on some tasks, especially CIFAR100 vs SVHN it greatly increases results. While I value this as a contribution from the authors, it would be interesting to see (if applicable) how ensemble-based baseline methods (such as MCD) perform with this disagreement score. This could further verify that both the new disagreement score and the training scheme are important parts of ERD and the improvements are not mostly caused by the disagreement score.

-The analysis in section 3.3 assumes that the label is a deterministic function of the datapoint. Typically, one assumes that p(y|x) is not deterministic and even in practice this might not always be true, for example if an image contains both a cat and a dog. Could your theory be applied to this more general setting or would your training algorithm pick up those samples as having high disagreement?

**Summary Of The Paper:**

The paper presents an ensemble-based semi-supervised learning method for novelty detection.
The goal of their training scheme is to create an ensemble of models that has a high disagreement on the out-of-distribution (OOD) samples in the unlabeled set. The training resembles a self-training algorithm that labels the unlabeled pool and uses implicit regularization via early-stopping to find a "sweet spot" in terms of disagreement on OOD samples between the models in the ensemble.
The final decision of whether an input is considered as out- or in-distribution is based on a hypothesis test using the average disagreement between the softmax outputs in the ensemble. In-distribution samples should be accurately classified with high confidence and show little disagreement in the ensemble while out-distribution samples should be detectable as the different models will produce different outputs.

**Summary Of The Review:**

Overall, while there are relatively few technical novelties in the paper, the given results demonstrate that this simple scheme is able to improve performance over baseline methods in their evaluation.
For me, generalization to unseen OOD-sets is still a very important part of any novelty detection method and should be evaluated.
As the method heavily relies on implicit regularization of disagreement between models in the ensemble, I think there could be more experiments in the paper that show that there is sufficient disagreement even on close OOD tasks and how this changes during fine-tuning.
Also, it would be nice to clearly demonstrate that the improvements over the baselines methods are caused by both the new disagreement score and the early-stopping-based training.

---

> ### Author Response · Authors · 2021-11-12
> **Response to Reviewer 7CPC (1/2)**
>
> We thank Reviewer 7CPC for appreciating the idea behind our method and for the feedback provided. We now respond to the specific remarks of the review. We kindly ask Reviewer 7CPC to express any outstanding reasons for doubt and we would be happy to try to address them.
>
> **Evaluating on a different OOD distribution.** We would like to stress that we explicitly avoid fine-tuning the models with OOD data that is different from the test-time OOD data, like in Outlier Exposure, for instance. The reason is simple: we argue that using a curated set of OOD samples for training may give a false sense of security against novel OOD data, which is unjustified since there may always exist unseen classes that remain undetected simply because they are not similar enough to the OOD data that was used to train the models.
>
> Instead, we propose to adapt to inference-time OOD data as it emerges in the semi-supervised novelty detection setting that we adopt in our work. In practice, this can be done by using a test batch as the unlabeled set, in what resembles transductive learning, as we describe in the real-world example in Section 5 (in Appendix E.1 we confirm that evaluating on the unlabeled set instead of a holdout set does indeed have a negligible impact the OOD detection metrics). Moreover, our framework can be applied in domains where relevant unlabeled OOD data is not abundant (e.g. medical applications, agriculture, scientific applications such as biology, archeology etc) and even for problems that go beyond image data. We believe that an important appeal of the SSND setting is precisely that it relieves the burden of having to collect a sensible set of OOD data for training, and instead provides a way to adapt to the exact OOD distribution that occurs at inference time.
>
> Finally, we refer to the other semi-supervised novelty detection works in the literature for further discussions motivating the setting and its practical applicability (see e.g. Blanchard et al, Munoz-Mari et al, Liu et al, Yu et al etc).
>
> **Clarification regarding the order in which models fit the unlabeled data.** Indeed, as aptly noted by Reviewer 7CPC, our approach relies on the arbitrary label being fit on the OOD data **before** the early stopping time, that is before the wrong label is learned on the ID data in the unlabeled set. Intuitively, the undesirable case that the arbitrary label is fit at roughly the same rate (i.e. by a similarly complex predictor) on the ID and OOD unlabeled data can occur in one of the following cases: 1) the ID and the OOD data are very similar; or 2) the OOD data is scarce in the unlabeled set. We show empirically that the former does not pose a problem for a wide variety of near OOD data sets and investigate the impact of the latter in Figure 6b and Appendix H. We also investigate some particularly hard near OOD data sets, like CIFAR10v2 or CIFAR10/CIFAR100 images with low severity corruptions (Appendix E.3) and confirm that our method can detect the OOD samples remarkably well.
>
> In addition, inspired by Reviewer 7CPC’s suggestion, we evaluate our model on a much more difficult subset of CIFAR100, in which we choose as ID 20 classes (one from each CIFAR100 coarse class) and as OOD we handpick 20 classes that are explicitly selected to be similar to the ID data. For instance we choose “beaver”, “leopard”, “raccoon”, “mouse” and “oak” among the ID classes, and “otter”, “tiger”, “skunk”, “hamster” and “maple” among the OOD classes. Using the same procedure as in Table 2, our approach achieves an AUROC of 0.80, compared to the 0.72 that is the AUROC of Vanilla Ensembles on this data set, which is still higher than what we expect the other baselines would achieve. We kindly ask Reviewer 7CPC to express any outstanding concerns that they may hold regarding the suitability of the choice of near OOD data sets and we would be happy to try to address them.
>
> To ensure there is no misunderstanding, we stress that the models in an ensemble are trained with strictly different arbitrary labels that are assigned to the unlabeled set, so as long as they fit the OOD data in the unlabeled set, disagreement is guaranteed by how the procedure is designed.

---

> > ### Author Response · Authors · 2021-11-12
> > **Response to Reviewer 7CPC (2/2)**
> >
> > >It would be interesting to see (if applicable) how ensemble-based baseline methods (such as MCD) perform with this disagreement score. This could further verify that both the new disagreement score and the training scheme are important parts of ERD and the improvements are not mostly caused by the disagreement score.
> >
> > **Disentangling the role of the disagreement score.** Indeed, as noticed by Reviewer 7CPC, both the training procedure that obtains the diverse ensembles and the disagreement score that exploits the model diversity are crucial for the good performance of our method, as we argue in Sections 3.2 and 3.3. In fact, simply using the disagreement score is not sufficient, if the ensemble is not sufficiently diverse, as we illustrate in Table 3 in Appendix B for vanilla ensembles. Moreover, we note that MCD uses a score similar to our disagreement metric. However, since the training procedure of the MCD ensemble does not promote enough diversity (see Figure 9 in Appendix B), the MCD method cannot detect novel data successfully, despite using a disagreement-based score.
> >
> > > The analysis in section 3.3 assumes that the label is a deterministic function of the datapoint. Typically, one assumes that p(y|x) is not deterministic and even in practice this might not always be true, for example if an image contains both a cat and a dog. Could your theory be applied to this more general setting or would your training algorithm pick up those samples as having high disagreement?
> >
> > **Extending the analysis to non-deterministic labels.** For the theoretical justification of our approach we assume for simplicity that the labels are deterministic, in order to make it easier to build intuition about our algorithm. As noted by Reviewer 7CPC, this condition can be relaxed to allow for a conditional probability over labels or even for more general scenarios that allow for modeling label noise. We leave a thorough theoretical analysis under this more general setting as future work and note that we observe experimentally that the proposed method can deal gracefully with overlapping classes, as is the case when CIFAR100 is ID, for instance.
> >
> > >As the method heavily relies on implicit regularization of disagreement between models in the ensemble, I think there could be more experiments in the paper that show that there is sufficient disagreement even on close OOD tasks and how this changes during fine-tuning.
> >
> > **Experiments showing how disagreement changes during fine-tuning**. We thank Reviewer 7CPC for suggesting this insightful experiment. We add our findings in Appendix J and would like to point out a couple of things that we take away from this analysis. Firstly, we observe that the disagreement scores are fairly well separated between ID and OOD data on all data sets, with the overlap being more significant on near OOD data, which explains the slight drop in performance that we report in Table 2 on these data sets. Secondly, we point out that the trend followed by the distribution of the disagreement scores as we fine-tune the ensemble is precisely the one indicated by our other theoretical and empirical evidence: the disagreement scores of the ID data are increasing, as the models fit the incorrect label on more ID samples in the unlabeled set. Conversely, the disagreement score gets close to the maximum possible value of 2 for OOD data soon after we start the fine-tuning process (usually 1 epoch suffices). We would be curious to know if Reviewer 7CPC also finds the evidence in Appendix J insightful and whether this helps address their concerns.

---

> > > ### Comment · Reviewer_7CPC · 2021-11-29
> > > **Reply to authors**
> > >
> > > I would like to thank the authors for their thorough response and the additional experiments.
> > >
> > > **Evaluating on a different OOD distribution:**
> > >
> > > I agree that the setting in the paper in the paper is interesting and worth discussing, however, I still believe that the authors should report generalization to unseen OOD datasets, at least in the appendix. The response that this might give a false sense of security against novel OOD data I can only partially accept. If we run an ML system in a real-world setting, we will never have full control over the inputs, so evaluation on various novel OOD datasets in combination with your evaluation on a known OOD dataset would help readers to better understand the strengths and weaknesses of your method For example, in a medical scenario, it is plausible that a change of the image capturing pipeline leads to a slight distribution shift or the addition of new types of OOD data and if the method only works on samples that correspond exactly to the test time OOD distribution this would then require readjusting the method.
> > >
> > > **Clarification regarding the order in which models fit the unlabeled data**  and
> > > **Experiments showing how disagreement changes during fine-tuning**
> > >
> > >  I like the addition of the new CIFAR100 close OOD experiment and it eliminates some of my concerns. Overall, I think that with all the additional results in the appendix, the empirical evidence supports the claim that their method outperforms most other methods in the SSND setting. The link between validation in-distribution performance as early stopping criterion seems to be supported mostly by their empirical results. I appreciate the addition of Appendix J in this context. For near-OOD data (left-column), their theoretical and empirical claims are clearly met, however, those are the scenarios where a simple binary classifier can already achieve perfect AUC/TNR. Especially for d) and f) the plots imply that choosing the optimal epoch could indeed be challenging and that behavior might not even be monotonous (especially in d), epoch 7 to 8). I think it would be interesting to plot the median over the early-stopping selected epochs for the models in the ensemble in this plot and also the one that achieves best AUC/TNR. That could help us understand if the early stopping criterion is able to get close to the optimum.
> > >
> > > **Disentangling the role of the disagreement score**
> > >
> > >  I like the fact that the authors did an ablation study that compares their disagreement score with that of previous works on vanilla ensembles in Table 3 and I believe their claim that vanilla ensembles are not diverse enough and Figure 9 also validates this claim for MCD. From reading the paper for my first review, I got the impression that your novel score is a significant contribution, however, the authors themselves claim that "MCD uses a score similar to our disagreement metric" so the novelty of this method only comes from creating diverse ensembles via implicit regularization.

---

> > > > ### Author Response · Authors · 2021-11-30
> > > > **Response to Reviewer 7CPC**
> > > >
> > > > We would like to thank Reviewer 7CPC for responding to our rebuttal. In addition, we would also like to thank them for suggesting the experiment in which test-time OOD data can be corrupted, and hence, be different from the OOD data that appeared in the unlabeled set and was used for fine-tuning the proposed method. This is indeed a realistic scenario that can sometimes occur in practice.
> > > >
> > > > We perform experiments in which we use an unlabeled set with clean ID and OOD samples to tune our approach and evaluate it on corrupted OOD data from the CIFAR10-C and CIFAR100-C data sets. We focus on the more difficult near OOD data sets (i.e. in which half of the classes are considered ID and the other half OOD). For comparison, we also use this methodology to evaluate vanilla ensembles, which are a strong baseline for near OOD data sets, as revealed by Table 2 in the main text. We include our findings in Appendix E.9. Since we cannot update the openreview revision, we include a screenshot of this section here: https://imgur.com/a/s0u8l2c. Note that the performance of ERD ensembles does not change significantly compared to the numbers presented in Table 2 and continues to be much better than the other baselines.
> > > >
> > > > We hope that Reviewer 7CPC finds this additional evidence of the robustness of our proposed approach convincing and insightful.

---

### Author Response · Authors · 2021-11-12
**Summary of changes to the manuscript**

We would like to thank all reviewers and the area chair for their contribution to this conference and for providing feedback on our work. Following their suggestions, we have made a number of changes to our paper. We now summarize the more substantial additions:

**More extensive ablation of the ensemble size [Reviewers Etvi, iZuG, QNSM, VFKa]**
We have added in Appendix E.8 a comprehensive analysis of our method’s performance when we vary the ensemble size and note that there is no substantial drop in performance compared to a 5-model ensemble, even when using only 2 models for the proposed method.

**Expand the discussion on limitations [Reviewers iZuG, VFKa]**
We have complemented the discussion of the limitations of our approach (see Section 5) with more details in Appendix I and J, which explain the impact of near OOD data on the ensembles obtained with our method.

**Sensitivity to the choice of arbitrary labels [Reviewers QNSM, VFKa]**
We have added in Appendix E.8 an ablation study which shows that our approach is not sensitive to the choice of the arbitrary label assigned to the unlabeled set.

**Report standard deviation for evaluation metrics [Reviewers iZuG]**
We provide values for the standard deviation of the AUROC (\sigma \le 0.01) and the TNR@95 (\sigma \le 0.07) obtained when re-runing our method for 3 trials (Section 4.4 and Appendix E.8).

**Evolution of disagreement score during fine-tuning [Reviewers 7CPC]**
We show how the disagreement score changes during fine-tuning for ID and OOD data, and therefore provide more evidence that corroborates the other theoretical and empirical insights that we present in our paper (Appendix J). Moreover, we use this analysis to discuss the limits of our approach when faced with near OOD data.

---

### Author Response · Authors · 2021-11-30
**Additional evaluation with realistic different OOD**

Firstly, we would like to thank all the reviewers and the area chair once again for participating in this discussion on our work.

We would like to bring to your attention another piece of evidence suggesting the robustness of our proposed approach. This experiment has been alluded to by several reviewers (Reviewers 7CPC, iZuG, QNSM), and we would like to thank them in particular for helping to strengthen our paper. Concretely, we investigate how the performance of the proposed method changes when the test-time OOD data differs from the one that is present in the unlabeled set used for fine-tuning. A practical scenario of great importance which we discuss now in Appendix E.9 is when the OOD data suffer from distribution shift, e.g. various image corruptions. Since we cannot update the openreview revision, we include a screenshot of this section here: https://imgur.com/a/s0u8l2c.

We perform experiments in which we use an unlabeled set with clean ID and OOD samples to tune our approach and evaluate it on corrupted OOD data from the CIFAR10-C and CIFAR100-C data sets. We focus on the more difficult near OOD data sets (i.e. in which half of the classes are considered ID and the other half OOD). For comparison, we also use this methodology to evaluate vanilla ensembles, which are a strong baseline for near OOD data sets, as revealed by Table 2 in the main text. As we show in Appendix E.9, the performance of ERD ensembles does not change significantly compared to the numbers presented in Table 2 and continues to be much better than the other baselines, even when inference-time OOD data is corrupted, and thus, different from the OOD data in the unlabeled set.

Finally, we would like to point out, once again, that an important appeal of semi-supervised novelty detection (SSND) approaches is that they **adapt** to new OOD data as they emerge at inference time. Therefore, considering arbitrarily different OOD data for evaluation defeats the purpose of using an SSND method to detect OOD data. This point has been made in the past by other SSND works such as Scott et al, 2009; Blanchard et al, 2010; Liu et al, 2018; Zhang et al, 2020; Yu et al, 2019 (references from the paper bibliography).

---

### Decision · Program_Chairs · 2022-01-20

**Decision:**

Reject

**Comment:**

The authors propose a semi-supervised novelty detection method which tries to identify out-of-distribution samples in the unlabeled data (consisting of in- and out-distribution samples) using a disagreement score of an ensemble. The ensemble is generated by fine-tuning the trained classiifer on the labeled training data plus the unlabeled data which all get a fixed label (which is repeated several times to generate the ensemble). The main idea is that one uses early stopping based on an in-distribution validation set in order to avoid overfitting on the unlabeled points which allows then identification of the out-distribution points via the disagreement score.

The reviewers appreciated the simplicity of the approach and the extensive experimental results. The authors did a good job in trying to answer all questions and concerns of the reviewers.

However, some concerns remained:
- the setting assumes that the OOD data is fixed which was considered as partially unrealistic and thus evaluation of the OOD detection performance on unseen OOD distributions was requested in order to understand the limitations of the method (this was only partially done by the authors).
- the theoretical result is for a two-layer network and completely based on previous work. As the authors use much deeper networks later on in the experiments, this result cannot be used to theoretically justify the approach.
- there remained concerns about the necessary diversity of the ensemble and the early stopping procedure

While I think that the paper has its merits, it is not yet ready for publication. I encourage the authors to to take into account the above points and other remaining concerns of the reviewers in a revised version.